# A conserved function of Human DLC3 and *Drosophila* Cv-c in testis development

Sol Sotillos[1]*, Isabel von der Decken[2‡], Ivan Domenech Mercadé[2‡], Sriraksha Srinivasan[3], Dmytro Sirokha[4], Ludmila Livshits[4], Stefano Vanni[3], Serge Nef[5], Anna Biason-Lauber[2]*, Daniel Rodríguez Gutiérrez[2†], James Castelli-Gair Hombría[1†]

[1]Centro Andaluz de Biología del Desarrollo, Seville, Spain; [2]Department of Endocrinology, Metabolism and Cardiovascular research, University of Fribourg, Fribourg, Switzerland; [3]Department of Biology, University of Fribourg, Fribourg, Switzerland; [4]Institute of Molecular Biology and Genetics, National Academy of Sciences of Ukraine, Kyiv, Ukraine; [5]Department of Genetic Medicine and Development, Faculty of Medicine, University of Geneva, Geneva, Switzerland

**\*For correspondence:**
ssotmar@upo.es (SS);
anna.lauber@unifr.ch (AB-L)

[†]These authors contributed equally to this work
[‡]These authors also contributed equally to this work

**Abstract** The identification of genes affecting gonad development is essential to understand the mechanisms causing Variations/Differences in Sex Development (DSD). Recently, a DLC3 mutation was associated with male gonadal dysgenesis in 46,XY DSD patients. We have studied the requirement of Cv-c, the *Drosophila* ortholog of DLC3, in *Drosophila* gonad development, as well as the functional capacity of DLC3 human variants to rescue *cv-c* gonad defects. We show that Cv-c is required to maintain testis integrity during fly development. We find that Cv-c and human DLC3 can perform the same function in fly embryos, as flies carrying wild type but not patient DLC3 variations can rescue gonadal dysgenesis, suggesting functional conservation. We also demonstrate that the StART domain mediates Cv-c's function in the male gonad independently from the GAP domain's activity. This work demonstrates a role for DLC3/Cv-c in male gonadogenesis and highlights a novel StART domain mediated function required to organize the gonadal mesoderm and maintain its interaction with the germ cells during testis development.

## Editor's evaluation

This important study demonstrates a conserved function of somatic human DCL3/ *Drosophila* Cv-c in the development of the testis. The evidence supporting the claims of the authors is compelling as they combine the human genetics of a patient with gonadal dysgenesis with genetic rescues in *Drosophila*. This work will be of interest to developmental biologists and human geneticists. A major strength of the paper is proving that the presumptive lipid-interacting StART domain (but not the GAP domain) of DCL3/Cv-c is required for testis development, which opens a new area of investigation.

## Introduction

Human sex development relies on the correct differentiation and function of the gonads. This requires a delicate functional balance between genes, cells, and hormones. Mutations affecting the determination and differentiation of the gonads can lead to Differences of Sex Development (DSD, also called Variations of Sex Development, VSD) where the female (XX) or male (XY) sex karyotype does not match with gonadal and anatomical development (*Cools et al., 2018*). The primary cause of nearly

50% of DSD cases remains unknown (*Barseghyan et al., 2018*; *Buonocore et al., 2019*), suggesting the existence of further sex-developmental mechanisms still awaiting discovery.

Gonadogenesis can be subdivided into three stages: specification of precursor germ cells, directional migration towards the somatic gonadal precursors, and gonad compaction. In mammals, somatic cells, i.e. Sertoli cells in male and granulosa cells in females, play a central role in sex determination with the germ cells differentiating into sperms or oocytes depending on their somatic mesoderm environment. In humans, primordial germ cells (PGCs) are formed near the allantois during gastrulation around the fourth *gestational week* (*GW*) and migrate to the genital ridge where they form the anlage necessary for gonadal development (GW5–6). Somatic mesodermal cells are required for both PGCs migration and the formation of a proper gonad. Once PGCs reach their destination, the gonadal cells join them (around GW7–8 in males, GW10 in females) and provide a suitable environment for survival and self-renewal until gamete differentiation (*Jemc, 2011*). Thus, mutations in genes regulating somatic Sertoli and granulosa support cell function in humans are often associated with complete or partial gonadal dysgenesis in both sexes and sex reversal in males (*Brunello and Rey, 2021*; *Knower et al., 2011*; *Zarkower and Murphy, 2021*). Other mesodermal cells, the Leydig cells, also play an important role in the testis by being the primary source of testosterone and other androgens and maintaining secondary sexual characteristics.

The central elements of gonadogenesis are relatively well conserved among species. In *Drosophila*, PGCs are formed in the blastoderm and are carried passively into the gut where they enter the embryo after crossing the intestinal epithelium. PGCs migrate towards their final position where they coalesce forming a compact gonad. The dependency of PGCs on somatic gonadal cells during development is also well conserved. In mouse mutants without a genital ridge, the PGCs can migrate but remain immature (reviewed in *Cooke and Moris, 2021*). In the fly, somatic gonadal cells can coalesce into a gonad in the absence of PGCs, but the PGCs are unable to coalesce in the absence of somatic cells (*Brookman et al., 1992*). Similarly, subsets of somatic gonadal cells produce steroid hormones (testosterone in mammals *Zirkin and Papadopoulos, 2018* and ecdysone in insects *Tajouri et al., 2018*), which have a global influence on the organism.

Abnormal gonadogenesis in human 46,XY individuals leads to under-masculinization, resulting in incomplete sexual characteristics at birth. Patients with complete gonadal dysgenesis present female external genitalia and hypogonadotropic hypogonadism with lack of secondary sex characteristics (*Rocha et al., 2011*). In mammalian models, gonadal dysgenesis causes infertility, DSD, and sex reversal while in insect models it leads to sterility.

Recently, the human X-linked *DLC3* gene (also known as *STARD8*) has been implicated in a case of 46,XY gonadal dysgenesis in two patients carrying a variant in the StART domain (*Ilaslan et al., 2018*). The Deleted in Liver Cancer (DLC) proteins belong to the RhoGAP family of small GTPase regulators. In vertebrates, there are three members (DLC1, 2, and 3) whereas *Drosophila* has a single orthologue, *crossveinless-c* (*cv-c*) (*Denholm et al., 2005*). This family of proteins shares different domains: besides the Rho GTPase Activating Protein domain (GAP), they present a protein–protein interacting Sterile Alpha Motif (SAM) at the N terminal end and a Steroidogenic Acute Regulatory protein (StAR)-related lipid transfer (StART) domain at the C terminal. StART domains have been shown in other proteins to be involved in lipid interaction, protein localization, and function (*Braun and Olayioye, 2015*; *Clark, 2020*).

We previously reported that DLC1 and DLC3 can functionally substitute for Cv-c in *Drosophila* (*Sotillos et al., 2018*) opening up the use of *Drosophila* as a system to analyse the requirement of DLC3/Cv-c proteins during male gonadogenesis. Here, using *Drosophila*, we demonstrate that the RhoGAP Cv-c and DLC3 proteins have a conserved role in male gonad formation mediated by the StART domain, confirming the suspected DLC3 involvement in human testicular organogenesis.

## Methods

### Patient analyses

The patient UKR05 carrying mutation R887C (Poltava region) was one from the cohort of 45 DSD patients from different regions of Ukraine. The patient was with complete gonadal dysgenesis, primary hypogonadism.

Blood samples of patient and parents as well as clinical data were obtained after informed consent. Genomic DNA from the blood samples of the proband and his parents was isolated by using the QIAmp DNA Kit (Qiagen, Hilden, Germany).

Cytogenetic studies (GTG-banding, FISH—probes CEP, LSI probes: Yp11.3—SRY; Yp11.1-q11.1—DYZ3; Yq12—DYZ1; CEP—DXZ1) of the patient showed a 46,XY karyotype with Y-specific sequences including SRY gene.

Whole exome sequencing (WES) was performed, and relevant variants were later validated with Sanger sequencing.

We identified a hemizygous missense mutation NM_001142503.2 c.2659 C>T (p.Arg887Cys VarSome/p.Arg807Cys UniProt) (rs766188656) in STARD8 gene. This variant has Minor allele frequency (MAF) = 0.0000251 and was not previously implicated in the pathogenesis of any disease.

## Molecular dynamics simulations

### Atomistic simulations to investigate conformational dynamics of the StART domain

The structure of the StART domain of human DLC3/STARD8 (Uniprot ID Q92502) was obtained from Alphafold (AF) (*Jumper et al., 2021*). A truncated structure of the domain comprising residues 838–1012 was used. The WT and S993N systems were set up using the CHARMM-GUI Solution Builder (*Jo et al., 2008*) with a cubic box of edge length of 7.3 nm. The systems were solvated with TIP3P water and ionized with 0.12 M of sodium and chloride ions. Three independent replicas of 100 ns each were simulated for each system using the GROMACS (*Van Der Spoel et al., 2005*) 2018.6 package and the CHARMM36m force field (*Huang et al., 2017*). Initial equilibration was carried out by performing energy minimization using the steepest descent algorithm, followed by a short NVT and NPT equilibration of 100 ps each with position restraints on the backbone atoms of the protein. Production runs were performed at 310 K using a velocity-rescale thermostat (*Bussi et al., 2007*), with separate temperature coupling for protein and solvent particles, and a coupling time constant of 0.1 ps. The first 10 ns of the production runs were not considered for analysis. The molecular dynamics (MD) integrator was used for the production runs, with a time step of 2 fs. The Parrinello–Rahman barostat (*Parrinello and Rahman, 1981*) was used to maintain the pressure at 1 bar, with an isotropic pressure coupling scheme, a compressibility of $4.5 \times 10^{-5}$ bar$^{-1}$ and a coupling time constant of 2.0 ps. Electrostatic interactions were evaluated using Particle Mesh Ewald (PME) with a Fourier spacing of 0.16 nm, a cut-off of 1.2 nm, and an interpolation order of 4. Van der Waals (VDW) interactions were switched to zero over 10–12 Å. Bonds involving hydrogen atoms were constrained using the LINCS algorithm. Periodic boundary conditions were used in all three dimensions.

The distance between the Cα atoms of the residues was computed every 100 ps and the distribution was obtained using Kernel Density Estimation (KDE). *Figure 1D* was rendered using Visual Molecular Dynamics (VMD) (*Humphrey et al., 1996*).

### Coarse-grained simulations to investigate binding of the StART domain to lipid bilayers

Lipid bilayers with lateral dimensions of 20 × 20 nm and composition of 70% 1,2-dioleoyl-sn-glycero-3-phosphocholine (DOPC), 30% 1,2-dioleoyl-sn-glycero-3-phospho-L-serine (DOPS) were built using the CHARMM-GUI Bilayer Builder for Martini (*Qi et al., 2015*). The bilayers were equilibrated according to the standard CHARMM-GUI six-step equilibration protocol. Water molecules and ions were removed from the system, and the protein of interest was placed away from the membrane, such that the initial minimum distance between any particle of the protein and any particle of the membrane was at least 3 nm. The orientation of the protein was such that its principal axes were aligned with the x, y, and z directions of the system, with the longer dimension of the protein along z. The setup was then solvated and ionized with 0.12 M of sodium and chloride ions to neutralize it as well as reproduce a physiological salt concentration.

Eight independent replicas of 3 µs each were simulated using the GROMACS (2018.x) (*Van Der Spoel et al., 2005*) package and the Martini 3 forcefield (*Souza et al., 2021*). Initial equilibration was carried out by performing energy minimization using the steepest descent algorithm, followed by a short MD run of 250 ps. Production runs were performed at 310 K with a velocity-rescale thermostat (*Bussi et al., 2007*), with separate temperature coupling for protein, bilayer, and solvent particles and

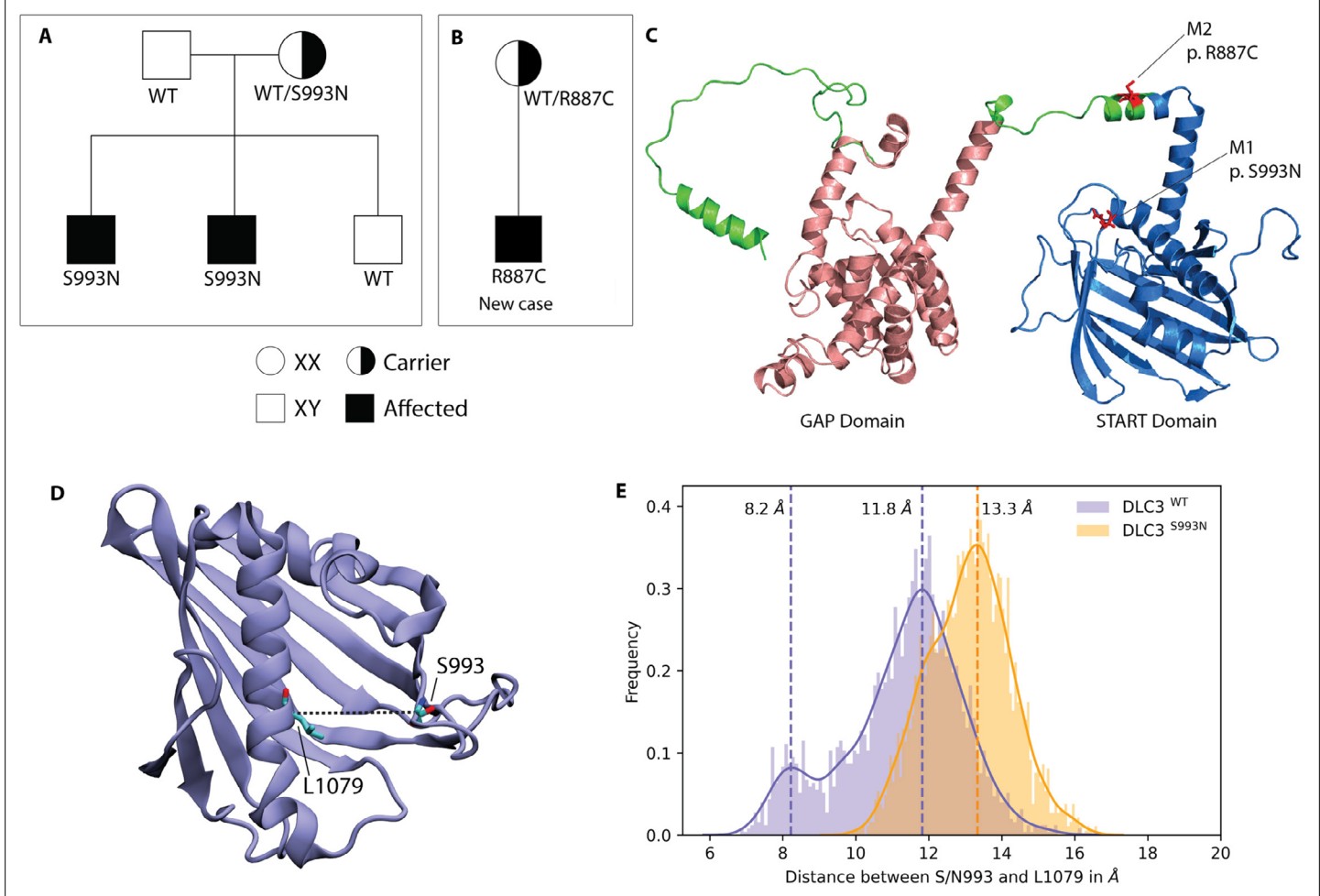

**Figure 1.** DLC3 variants associated to Differences of Sex Development (DSD) patients. (**A, B**) Family diagrams showing the segregation of two different DLC3 alleles (panel A is modified from *Ilaslan et al., 2018*). (**C**) Structure of DLC3 GAP and StART domains showing the localization of the p.R887C and the p.S993N mutations. The protein is shown in cartoon representation, with the GAP domain represented in orange and the StART domain in blue. (**D**) Structure of the StART domain of human DLC3/StARD8. The distance between the Cα atoms of residues S/N993 and L1079 (shown in licorice representation) was used to determine the open and closed transitions arising from the motion of the Ω1 loop. (**E**) Distribution of the distance between the Cα atoms of S/N993 and L1079 in atomistic simulations of the WT and S993N systems.

The online version of this article includes the following figure supplement(s) for figure 1:

**Figure supplement 1.** Binding of DLC3-StART domain to lipid bilayers.

a coupling time constant of 1.0 ps. The Parrinello–Rahman barostat (*Parrinello and Rahman, 1981*) was used to maintain the pressure at 1 bar, with a semi-isotropic pressure coupling scheme and a coupling time constant of 12.0 ps. The MD integrator was used for the production runs, with a time step of 20 fs. The Coulombic terms were calculated using reaction field (*Tironi et al., 1995*) and a cut-off distance of 1.1 nm. A cut-off scheme was used for the VDW terms, with a cut-off distance of 1.1 nm and Verlet cut-off scheme for the potential shift (*Verlet, 1967*). The Verlet neighbour search algorithm (*Páll and Hess, 2013*) was used to update the neighbour list every 20 steps with a buffer tolerance of 0.005 kJ mol$^{-1}$ ps$^{-1}$. Periodic boundary conditions were used in all three dimensions. The system setup and simulation parameters are in line with the recently proposed protocol for studying transient protein–membrane interactions with the Martini force field (*Srinivasan et al., 2021*).

Membrane-binding events were assessed using the time-trace of the minimum distance between the protein and the membrane, computed with the gmx mindist tool in GROMACS. Membrane-interacting residues were computed every 500 ps using an in-house script with the following protocol: a residue was considered to interact with the membrane if the distance between any bead of the residue and any lipid bead was lower than or equal to 0.5 nm. For each residue, we counted the

instances of residue–membrane interaction during the trajectory, summed this value over all the replicas, and computed a corresponding normalized value. *Figure 1D* and *Figure 1—figure supplement 1A, D* were rendered using VMD (*Humphrey et al., 1996*).

## Fly strains

The following lines: *Mi{PT-GFSTF.0}cv-cMI00245-GFSTF.0*, *Sxl::GFP* and *Df(3R)Exel 6267* were obtained from the Bloomington *Drosophila* Stock Center. The Cv-c::GFP fusion protein results from the integration into *cv-c* of a *gfp* sequence flanked by acceptor/donor splicing sequences present in a transgene (*Mi{MIC}t cv-c MI00245*) inserted into the third intron of the endogenous gene (*Venken et al., 2011*). *cv-c*[M62], *cv-c*[C524], and *cv-c*[7] were described in *Denholm et al., 2005*. *UAS-cv-c*[WT], *UAS-cv-c*[GAPmut], *UAS-cv-c*[ΔStART], and *UAS-Myc-DLC3*[WT] were described in *Sotillos et al., 2018*. *nos-nod::GFP* line is a gift from A. González Reyes. *c587-Gal4* line is a gift from E. Matunis. *six4-moe::GFP* line is described in *Sano et al., 2012* and *Vasa::GFP* line is a gift from P. Lasko. *Perlecan::GFP* was described in *Morin et al., 2001*. *Rho1*[72R] was described in *Strutt et al., 1997*. *twist-Gal4* was described in *Greig and Akam, 1993*.

Rescue experiments were performed crossing *twi-Gal4/CyO wg-lacZ; cv-c*[C524]*/TM6B risk-lacZ* females with *UAS-X/CyO wg-lacZ; cv-c*[C524]*/TM6B risk-lacZ* males or *c587-Gal4; cv-cC524/TM6B risk-lacZ* females with *UAS-X/CyO wg-lacZ; cv-c*[C524]*/TM6B risk-lacZ* males (where X corresponds to either *cv-c*[WT]; *cv-c*[GAPmut]; *cv-c*[ΔStART]; *Myc-DLC3*[WT] or *Myc-DLC3*[S993N]).

Flies were raised at 25°C.

## Constructs

To generate *UAS-Myc-DLC3*[S993N] a XhoI DLC3[WT] fragment was subcloned into pBlueScript (DLC3-Xho-pBs). DLC3-Xho-pBs was used as a template to mutate Ser at position 993 to Asn of human DLC3 (CCDS48134.1) using Pfu Polymerase.

The following primers were used:

>     Forward: 5'-TGTACCACTATGTCACCGACA-A-CATGGCACC-3'
>     Reverse: 5'-TGGGGTGCCATG-T-TGTCGGTGACATAGTG-3'

After PCR reaction, DNA was incubated with DpnI during an hour at 37°C to digest the methylated template DNA and transformed. Clones were sequenced by standard methods.

From the selected clone (DLC3[S993N]-Xho-pBs), a BglII fragment containing the mutation was substituted in the pUASt-Myc-DLC3[WT] to obtain pUASt-Myc-DLC3[S993N].

Constructs were injected by the *Drosophila* Consolider-Ingenio 2007 transformation platform (Spain).

We will share the flies or plasmids upon request to the corresponding authors.

## Immunohistochemistry

Embryos were collected on apple juice agar plates that contained yeast paste. For immunostaining experiments, female flies were allowed to lay eggs overnight onto plates at 25°C. Embryos were dechorionated 2 min in commercial bleach diluted in water (1:1), washed and fixed for 20 min in a phosphate-buffered saline (PBS)-formaldehyde 4%/heptane mix. After removing the fixative, methanol was added and shaken vigorously for 1 min to remove the vitelline membrane. After allowing both phases separate, sinking embryos were recovered, washed in clean methanol and rehydrated in PBS-Tween 0.1% (PBT) and preadbsorbed during an hour in PBT–1% bovine serum albumin (BSA) at room temperature (RT). Primary antibodies were used at the described concentration, diluted in PBT–1% BSA and incubated overnight at 4°C. Primary antibodies were washed 2 × 5 min in PBT and preadbsorbed 1 hr in PBT–1% BSA at RT. Embryos were incubated with the secondary antibody diluted at 1:400 in PBT–1% BSA at RT during 4 hr. After incubation embryos were washed 4 × 15 min in PBT–1% BSA at RT followed by 2 rinses in PBS before mounting in Vetacshield.

The following primary antibodies from the Developmental Studies Hybridoma Bank (DSHB) were used: rat anti-Vasa 1:20 (DSHB, VASA), mouse anti-Nrt 1:100 (DSHB, BP106), mouse anti-β-catenin 1:50 (DSHB, N27A1), rat anti-DE-cad 1:50 (DSHB, DCAD2), mouse anti-Eya 1:100 (DSHB, Eya), guinea pig anti-Ems 1:5.000 (gift from U. Walldorf, *Walldorf and Gehring, 1992*), rabbit-anti-Sox100B 1:1.000 (gift from S. Russell, *Nanda et al., 2009*), guinea pig anti-Tj 1:1000 (*Gunawan et al., 2013*),

rabbit anti-Perlecan 1:1.000 (gift from A. González-Reyes, *Díaz-Torres et al., 2021*), mouse anti-βgal 1:1.000 (Promega, Z378A), rabbit anti-GFP 1:300 (Invitrogen, A11122), and chicken anti-GFP 1:500 (Abcam, ab13970).

Secondary antibodies were coupled to Alexa488, Alexa555, or Alexa647 (Molecular Probes). Filamentous actin was stained with rhodamine-phalloidin (Molecular Probes, R415).

Fluorescent in situ hybridization was performed according to standard protocols adding a secondary fluorescent antibody (anti-goat Alexa555, Invitrogen A-21432); *cv-c* riboprobe was marked using DIG RNA Labelling Kit (Roche, 11 175 025 910). Images were taken on an SPE Leica confocal microscope and processed using FIJI and Adobe Photoshop programs.

### In vivo gonad analysis

In the *nos-nod::GFP* construct a nanos enhancer drives expression in the germ cells of a microtubule-binding GFP protein due to its fusion to the Nod protein fragment. In *P-Dsix4-eGFP::Moesin* construct a *six* enhancer drives expression in the male-specific somatic cells of the Moesin actin-binding domain fused to GFP (gift from S. DiNardo, *Sano et al., 2012*).

Embryos were dechorionated in bleach and positioned dorsally on top of a coverslip thinly coated with heptane glue and covered with a drop of halocarbon oil. Embryos were imaged on an SP5 Leica confocal microscope, using a ×40 oil immersion objective. For each movie, 26–30 time points were collected. For each time point, between 20 and 40 Z sections were collected (spaced between 0.5 and 1 µm). Movies were assembled using IMARIS and Fiji ImageJ software.

### Statistical analysis

Data were obtained from at least three biological replicas. Replica samples for each genotype were collected in parallel on different days. Stainings with anti-Vasa, anti-E-Cad, and anti-Ems were carried-out to distinguish the GCs, the gonad contour, and the testis, respectively. We captured confocal z-stacks (at 0.5-µm intervals) encompassing the entire width of testes (typically 40–45 slices). A testis was considered abnormal if one or more germ cells were partially or totally outside the gonad. Data were analysed by Microsoft Excel and GraphPad Prism. Statistical analyses were performed using Fisher test and standard error as described in *Xu et al., 2010*. Statistical significance was assumed by $p < 0.05$ (source data 1).

## Results

### The S993N mutation is located in a functionally important region of the DLC3-StART domain and can alter its conformational dynamics

The extremely rare (MAF = 0.00016 according to Gnomad) *DLC3* mutation was previously observed in two related DSD patients where a Serine (S) amino acid (aa) was substituted by an Asparagine (N) at position 993. The mutation was present in the heterozygous carrier mother (*Figure 1A*) and in the two 46,XY dysgenic patients but not in their 46,XY healthy sibling (*Ilaslan et al., 2018*). However, despite this strong correlation, no direct evidence linking DLC3 to gonadal dysgenesis was provided.

Recently, we have identified a third DSD patient carrying a DLC3 variant, who inherited the R887C extremely rare mutation (MAF = 0.00025) from the heterozygous mother (*Figure 1B*). The patient presented 46,XY gonadal dysgenesis, where the gonadectomies found on the right gonad connective tissue of the fallopian tube with thickened walls and partially obliterated lumen and, in some areas, fragments of ovarian tissue with sclerosed theca tissue and on the left gonad degenerative altered tissue of the testicle with areas of mucous and degeneration of tubules without signs of spermatogenesis. As this discovery reinforces the suspected involvement of DLC3 as a DSD factor, we decided to search for experimental evidence.

As a first approach we concentrated on the DLC3[S993N] variant, which localizes to the conserved StART domain, analysing in silico if the aa substitution modifies the protein structure. The Alpha-Fold (AF) model of DLC3 indicates that residue S993 is located in the Ω1 loop of the StART domain (*Figure 1C*), a loop that has been shown to be functionally important for several StART domains (*Gatta et al., 2018*; *Horenkamp et al., 2018*; *Iaea et al., 2015*; *Khelashvili et al., 2019*; *Naito et al., 2019*). To analyse if the S993N mutation affects the conformational dynamics of DLC3, we performed atomistic simulations of wild-type DLC3 and of the mutant protein DLC3-S993N in

water. We observed that the mutation causes a non-negligible effect on the conformation of the Ω1 loop that is located at the entrance of the hydrophobic cavity of the StART domain (*Figure 1D, E*). The simulations of the wild-type DLC3-StART domain show a transition between a 'closed' state, where the distance between the Cα atoms of S993 and L1079 located on the opposite C-terminal helix is 8.2 Å, and an 'open' state with an aperture of 11.8 Å. In contrast, the DLC3-S993N-StART domain showed a single 'open' conformational state where the binding pocket has an aperture of 13.3 Å.

We also performed MD simulations at the coarse-grain resolution level to predict membrane-interacting regions of the DLC3-StART domain (*Figure 1—figure supplement 1*). The StART domain was initially positioned at least 3 nm away from a bilayer of composition DOPC:DOPS (70:30) (*Figure 1—figure supplement 1A*), and eight independent replicas of the system were simulated for 3 μs each. We observed multiple binding and unbinding events between the protein and the bilayer in all replicas, as indicated by the minimum distance between them (*Figure 1—figure supplement 1B*). When the frequency of interaction for each residue of the protein with the bilayer was determined (*Figure 1—figure supplement 1C*), the N-terminus of the domain and the Ω1 loop (*Figure 1—figure supplement 1D*) show the highest frequency of interaction.

These results indicate an effect of the S993N mutation on the conformational dynamics of the DLC3-StART domain and suggest that alterations of this Ω1 loop could impair the domain's membrane interaction.

## Expression of the *cv-c RhoGAP* gene in the *Drosophila* gonads

Structural analyses have shown that DLC3 and Cv-c are highly conserved (*Figure 2A, B*) and that, in *Drosophila*, DLC3 can functionally substitute for Cv-c (*Sotillos et al., 2018*). However, previous studies of Cv-c have concentrated in ectodermal derived tissues despite *cv-c* being broadly expressed in the mesoderm (*Denholm et al., 2005*; *Sotillos et al., 2013*; *Sotillos et al., 2018*). To find out if *cv-c* RNA is specifically transcribed in the gonadal mesoderm we performed fluorescent RNA in situ hybridization in embryos double stained with antibodies to detect gonad-specific antigens. We observed that *cv-c* is transcribed in the testis mesoderm cells including the somatic gonadal cells ensheathing the germ cells, the male-specific somatic gonad precursors (msSGPs) located at the posterior of the gonad and the pigment cell precursors surrounding the whole testis (*Figure 2C, D*). We confirmed Cv-c translation in these cells analysing the expression of a Cv-c::GFP fusion protein expressed under the endogenous *cv-c* regulatory elements (*Figure 2F, H*). This Cv-c::GFP fusion protein has been shown to have identical distribution to the RNA expression (*Sotillos et al., 2018*). We did not observe comparable levels of *cv-c* mRNA nor Cv-c::GFP protein in the female gonad mesoderm (*Figure 2E, G, I*) nor in the germ cells of any sex, indicating a male-specific regulation of *cv-c* expression in the testis mesoderm.

As JAK/STAT signalling is required for GC and mesodermal cell development specifically during early male gonad formation (*Sheng et al., 2009*; *Sinden et al., 2012*) we wondered if *cv-c* expression in the testis mesodermal cells depends on JAK/STAT activity. To test possible interactions between Cv-c and JAK/STAT signalling we analysed if in *Df(1)os1A* mutant embryos, that lack all three Upd *Drosophila* JAK/STAT ligands thus rendering an inactive pathway (*Hombría et al., 2005*), the expression of Cv-c::GFP is affected. *Df(1)os1A/+; cv-c::GFP/+* females were crossed to *cv-c::GFP/+* males allowing us to compare *cv-c::GFP* expression in *Df(1)os1A* male embryos with their control siblings (*Figure 2—figure supplement 1A, B*). As previously described, in *Df(1)os1A* embryos the testis are smaller due to the lack of JAK/STAT activation, however we detect normal levels of Cv-c::GFP expression in the testis gonadal mesoderm.

We also analysed if Cv-c controls the activity of the JAK/STAT pathway (*Figure 2—figure supplement 1C–F*). Using an anti-STAT antibody (*Flaherty et al., 2010*) we first studied if STAT's nuclear accumulation is altered in *cv-c^C524* mutant embryos. Then, by using the 10XSTAT-GFP reporter which reveals the pathway's activation (*Bach et al., 2007*) we analysed if Cv-c controls JAK/STAT signalling. We did not find any difference between heterozygous and homozygous *cv-c^C524* testes in any case, indicating that the *JAK/STAT* and *cv-c* mutant phenotypes act independently of each other (*Figure 2—figure supplement 1C–F*).

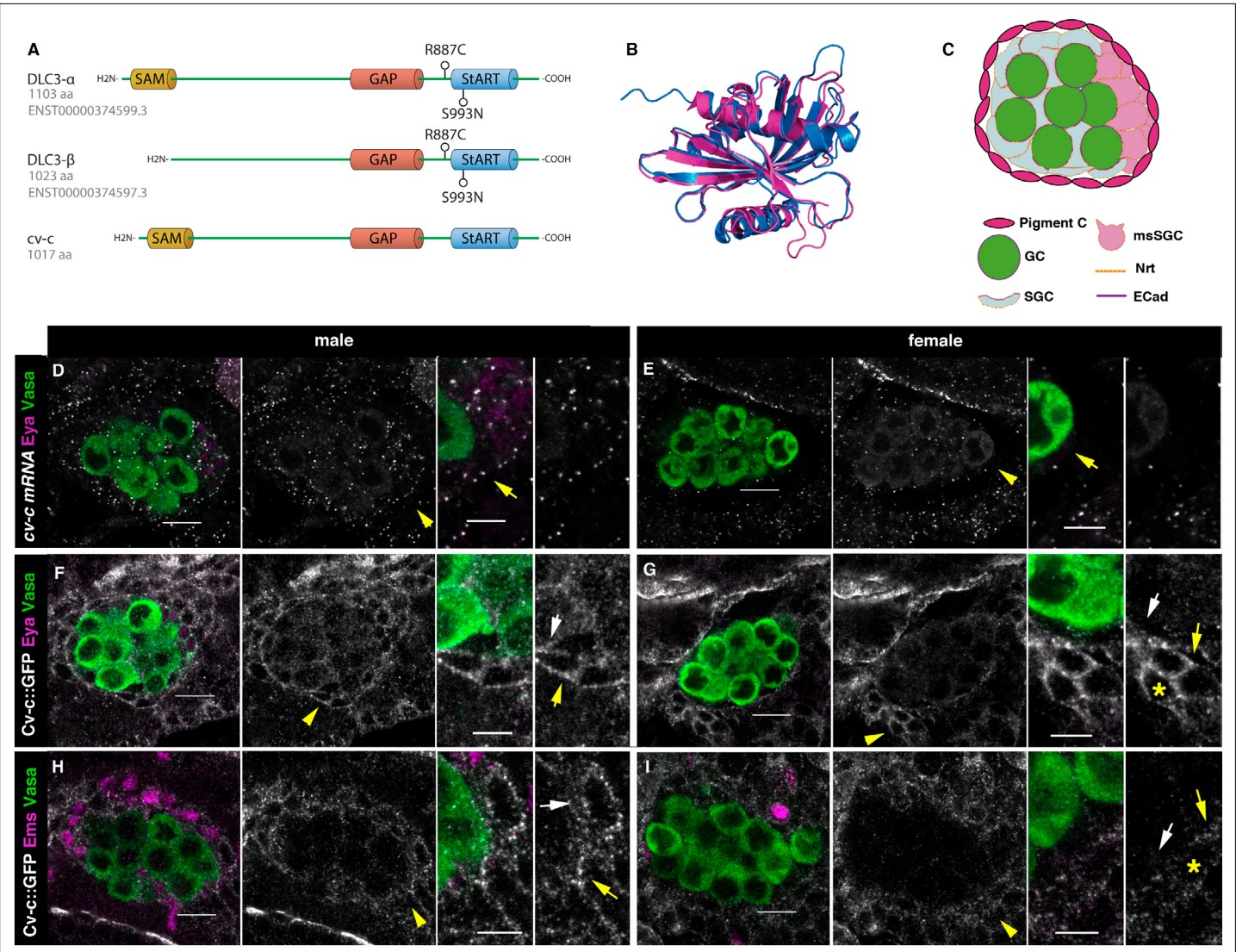

**Figure 2.** Cv-c and DLC3 structure and *cv-c* expression in the *Drosophila* gonad mesoderm. (**A**) Linear representation of the DLC3-α, DLC3-β, and Cv-c proteins. The SAM domain is represented in yellow, the GAP domain in orange, and the StART domain in blue. (**B**) Alignment of the DLC3 (blue) and Cv-c (magenta) StART domains. (**C**) Schematic representation of the cell types in a *Drosophila* testis at st17. Germ cells, green; pigment cells, magenta; somatic gonadal cells, grey; male-specific somatic gonadal cells, pink. RNA in situ hybridization of male (**D**) and female (**E**) st17 embryos shows general transcription of *cv-c* in the mesoderm. The right panels in D–I are close ups of the arrowed region in the central panels. (**D**) In the testis, comparable levels of mRNA puncta can be detected in the somatic mesoderm and in the gonadal mesoderm cells surrounding the male germ cells as are clearly observed in the male-specific somatic gonad precursors (msSGPs) marked by Eya (magenta, indicated by an arrowhead in grey panels and an arrow in the close up). (**E**) In the ovary, marginal levels of *cv-c* mRNA expression are observed in the gonadal mesodermal cells, creating a halo of decreased number of puncta surrounding the female germ cells contrasting with the *cv-c* expressing adjacent somatic mesodermal cells (arrow in close up). (**F–I**) Cv-c::GFP protein expression in male and female embryos. (**F, H**) In the testis Cv-c::GFP is detected in the gonadal mesoderm surrounding the germ cells including the male msSGPs (Eya, magenta F) and the pigment cell precursors (Ems, magenta H). (**G, I**) In females, no substantial GFP signal is detected in the gonadal mesoderm surrounding the germ cells. Note in (**F, H**) that Cv-c::GFP signal in the gonad mesoderm cells allows tracing the testis contour, while in ovaries (**G, I**) this is not possible. Higher levels of Cv-c::GFP are present in the ectodermally derived trachea and hindgut. In close ups white arrows point to membranes close to the germ cells, yellow arrows to the membrane of gonad mesodermal cells. In males, Cv-c::GFP can be detected in the membranes between gonadal and somatic mesodermal cells (**F, H**) whilst in females GFP can only be detected outside the ovary in the membrane of the somatic mesoderm (G, I asterisks). Scale bar: 10 and 5 µm in close ups.

The online version of this article includes the following figure supplement(s) for figure 2:

**Figure supplement 1.** JAK/STAT pathway and cv-c-independent functions.

## Cv-c functional requirement during male gonadogenesis in *Drosophila*

To test the functional significance of Cv-c expression in the fly gonads we analysed embryos homozygous for the lethal nonsense *cv-c*^M62 and *cv-c*^C524 alleles where stop codons result in truncated proteins lacking the GAP and StART domains (*Denholm et al., 2005*). We did not detect any

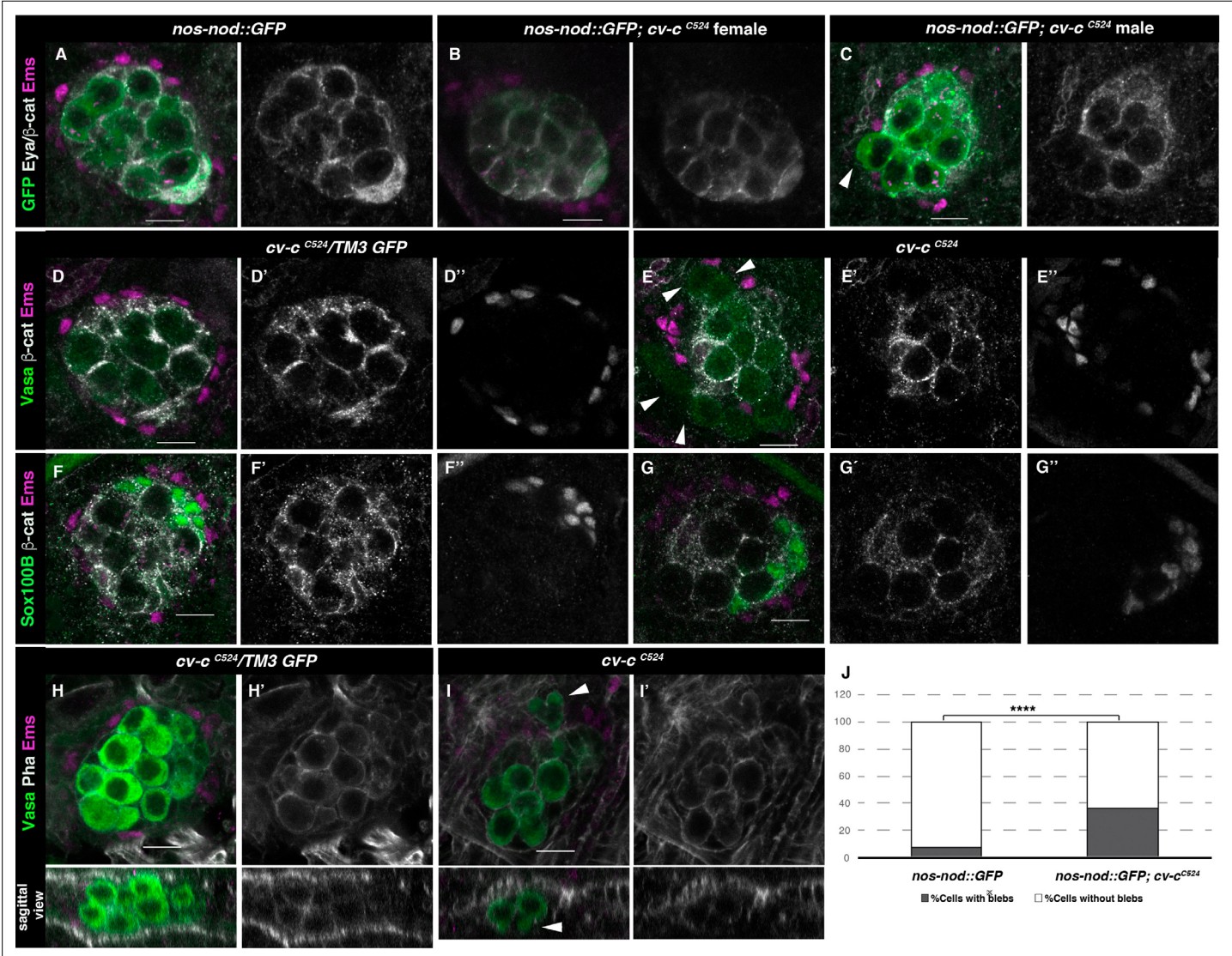

**Figure 3.** Gonad morphology in *cv-c* mutant embryos. (**A–C**) Gonads with germ cells labelled with *nos-nod::GFP* (green), pigment cells with anti-Ems (magenta), and male-specific somatic gonad precursor (msSGP) with anti-Eya (nuclear grey staining) and the AJs with anti-β-catenin (grey membranes). Right panels show Eya and β-catenin channel. (**A**) In the control testis, germ cells are ensheathed by thin mesoderm extensions produced by the interstitial cells detectable by β-catenin staining. Similar germ cell ensheathment is observed in *cv-c*^C524 ovaries (**B**), while in *cv-c*^C524 testis (**C**) some germ cells become extruded from the gonad and are not enveloped by β-catenin (arrowhead). (**D–G**) Testes labelled with mesodermal specific markers to detect the pigment cells (Ems, magenta D–G) or the msSGPs (Sox100B, green F, G) in heterozygous (**D, F**) or *cv-c*^C524 homozygous mutant embryos (**E, G**). Grey channels in right panels correspond to β-catenin in (**D-G**), Ems in (**D, E**), or Sox100B in (**F, G**). All mesoderm cell types are specified in *cv-c* mutant testis despite morphological aberrations resulting in the pigment cell layer's discontinuity (compare D and F with E and G). (**H–J**) Testes stained with anti-Vasa (green) to label the germ cells and phalloidin (grey and right panels) to show actin filaments in heterozygous (**H**) or homozygous *cv-c*^C524 mutants (**I**). Germ cells in mutant testes present protrusions compatible with migratory movements (arrowheads). Z sections are shown below H, I panels. Scale bar: 10 µm. (**J**) Quantification of blebbing cells in wild-type or mutant *cv-c* background using Fisher test; ****p value <2.2e−16 (*nos::GFP* N = 575 and *nos::GFP;cv-c*^C524 N = 744) (*Figure 3—source data 1*).

The online version of this article includes the following source data and figure supplement(s) for figure 3:

**Source data 1.** Raw data.

**Figure supplement 1.** Actin cystoskeleton in *cv-c* mutants testes.

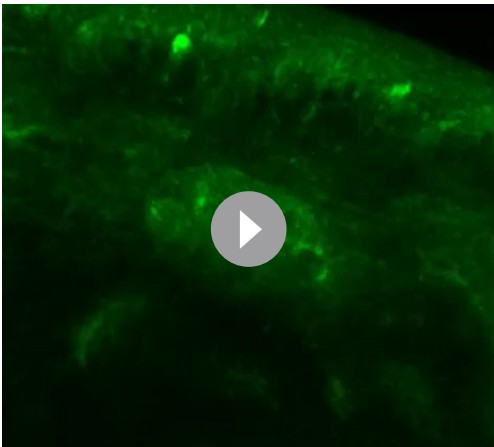

**Video 1.** In vivo gonad coalescence of a control heterozygous *cv-c*[C524]/+ testis. The germ cells are labelled with *nos-nod::GFP* and the male-specific somatic gonadal precursors with *six-moe::GFP*. Movie taken from st14 (prior to gonad coalescence) up to st16 (after coalescence).

https://elifesciences.org/articles/82343/figures#video1

**Video 2.** In vivo gonad coalescence of a homozygous *cv-c*[C524] testis. The germ cells are labelled with *nos-nod::GFP* and the male-specific somatic gonadal precursors with *six-moe::GFP*. Movie runs from st14 (prior to coalescence) to st16 (after coalescence) when germ cell extrusion becomes noticeable (arrow). Note that the early stages of testis development are normal until the germ cell extrusion begins at later stages.

https://elifesciences.org/articles/82343/figures#video2

major morphological defects in female gonads, confirming *cv-c* is not required for embryonic ovary development (*Figure 3B*). In contrast, homozygous or hemizygous *cv-c* mutant male embryos have abnormal testes containing germ cells that are not surrounded by the gonadal mesoderm (*Figure 3C and E* arrowheads compare with A and D, respectively). This defect is unlikely to be due to the abnormal specification of the gonad mesoderm cells, as using mesodermal specific markers we can observe the presence of all cell types (*Figure 3C, E, G*). However, the gonad mesoderm cells are frequently displaced, with the pigment cells failing to completely surround the mutant testis (*Figure 3E*).

To test if these defects are due to the abnormal migration of the germ cells or the mesoderm gonadal precursors during early gonad organogenesis [up to gonad coalescence at stage 15 (st15)] or to later defects on testis maintenance, we labelled the germ cells using *nos-nod::GFP* and the mesodermal cells with *six-moe::GFP* to investigate gonad formation *in vivo*. Using this setup, we could observe how the migrating germ cells and the somatic cells converge during normal development, coalescing at embryonic st15 to form a stable spherical gonad (*Video 1*). Analysis of *cv-c*[C524] homozygous mutant embryos in the same conditions revealed that development is normal up to st15, with the testis compacting into a spherical gonad (*Video 2*). However, after this stage the germ cells become extruded from the testis (*Videos 2 and 3*). Analysis of fixed mutant testes shows that the extruded germ cells extend blebs that are more characteristic of the earlier migratory phase (*Figure 3I*, arrowheads and J; *Figure 3—source data 1*; *Jaglarz and Howard, 1995*). These blebs are not observed in

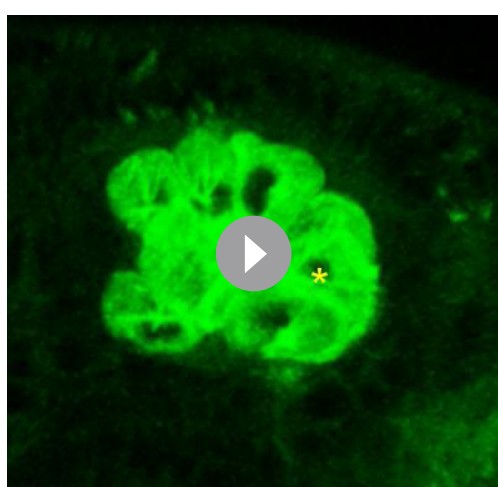

**Video 3.** Selected planes from the testis presented in *Video 2* to show more clearly the extrusion of an internal germ cell (asterisk).

https://elifesciences.org/articles/82343/figures#video3

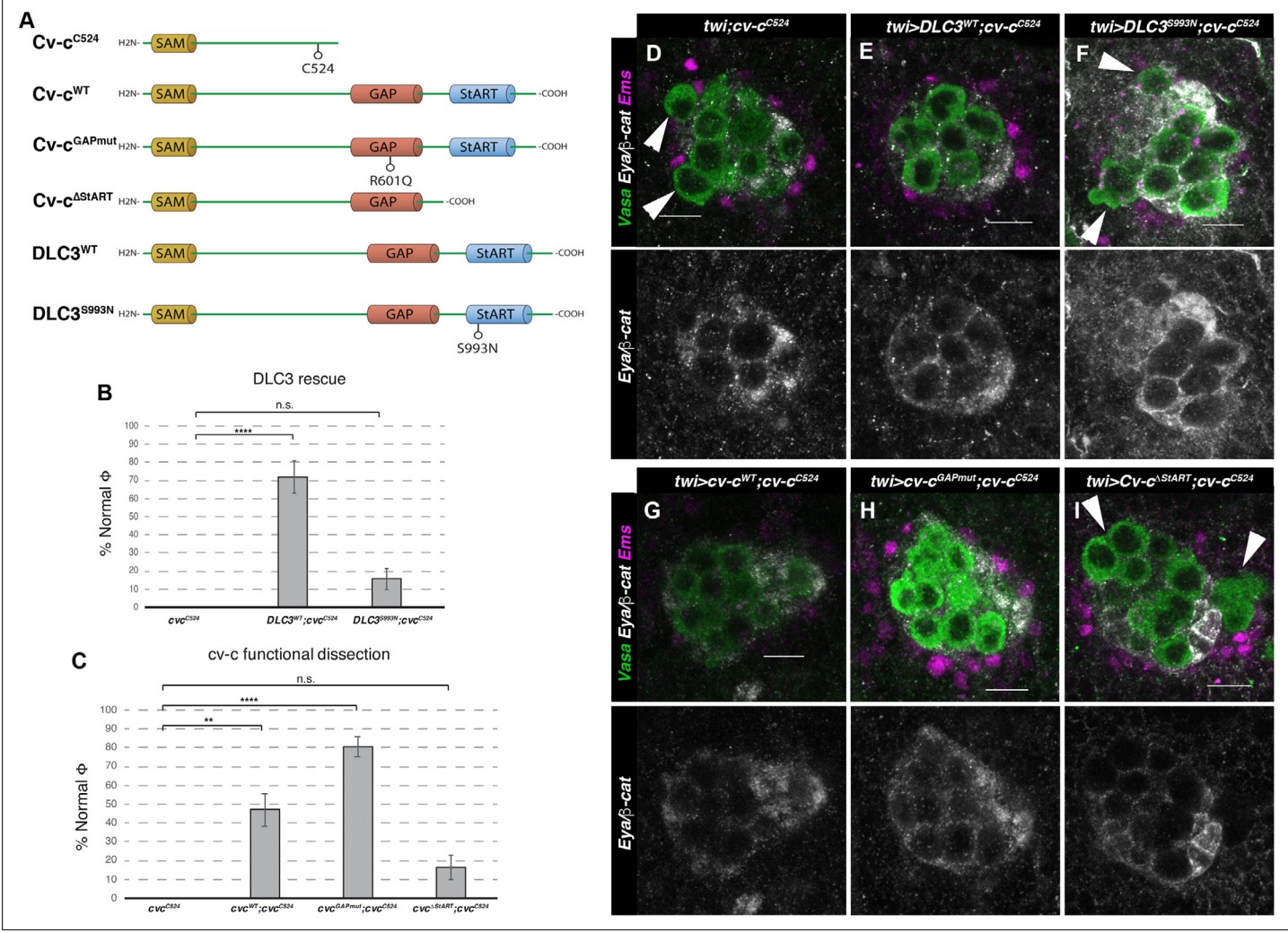

**Figure 4.** Rescue of *cv-c* mutant testes. (**A**) Schematic representation of Cv-c and DLC3 protein variants studied. Rescue of the dysgenic testis of *cv-c*^C524^ homozygous mutant males after expressing the specified (**B**) DLC3 or (**C**) Cv-c protein variants under UAS control with the pan mesodermal *twi-Gal4* line. Phenotypic rescue is shown as percentage of testes where all germ cells are encapsulated inside the testis. Representative images of testes in (**D**) control homozygous *cv-c*^C524^ animals, or homozygous *cv-c*^C524^ animals expressing in the mesoderm either (**E**) *UAS-DLC3*^WT^, (**F**) *UAS-DLC3*^S993N^, (**G**) *UAS-Cv-c*^WT^, (**H**) *UAS-Cv-c*^GAPmut^, or (**I**) *UAS-Cv-c*^ΔStART^. Arrows in D, F, I point to extruded germ cells that are not surrounded by β-catenin. Testes are stained with anti-Vasa to label the germ cells (green), anti-Ems to label the pigment cells (purple), and anti-Eya and anti-β-catenin to label the male-specific somatic gonad precursors (msSGPs) and the membranes ensheathing the germ cells, respectively (grey in lower panels). Scale bar: 10 μm. Fisher test, *Cv-c*^WT^ p = 0.0017 (N = 34), *Cv-c*^GAPmut^ p < 0.0001 (N = 56), *DLC3*^WT^ p < 0.0001 (N = 28), *Cv-c*^ΔStART^ p = 0.3005 (N = 31), and *DLC3*^S993N^ p = 0.3180 (N = 38) (ns, p > 0.05; **p < 0.001; ****p < 0.0001) (*Figure 4—source data 1*).

The online version of this article includes the following source data for figure 4:

**Source data 1.** Raw data.

the ovaries of *cv-c* mutant females (*Figure 3—figure supplement 1B*) nor in the wild-type testis after gonad compaction (*Figure 3H, J*).

## Rescue of *cv-c Drosophila* testis defects with human DLC3

We have previously shown that DLC3 can rescue the mutant phenotypes caused by *cv-c* mutations in the Malpighian tubules, the kidney-like structures of the fly, indicating these homologous *Drosophila* and human proteins conserve similar functions (*Sotillos et al., 2018*). Therefore, given that *cv-c* is expressed and required in the male *Drosophila* gonad, and that DLC3 can functionally substitute for Cv-c in some tissues, we investigated if DLC3 is also capable of rescuing the mutant gonadal defects observed in *cv-c*^C524^ homozygous embryos (*Figure 4A, D*). Using the UAS/Gal4 system to express

wild-type DLC3 protein with the pan mesodermal *twi-Gal4* driver line in otherwise *cv-c^C524* mutant embryos, we efficiently rescued the testis defects (**Figure 4A, B, E**) pointing out to the conservation of DLC3/Cv-c function in gonadogenesis. In contrast, expression in the same conditions of the DLC3^S993N StART protein present in human patients is not capable of efficiently rescuing the testis phenotype (**Figure 4A, B, F**).

## Testis development in *Drosophila* requires the StART domain

To elucidate the molecular mechanisms mediating DLC3/Cv-c function in gonad development, we analysed the capacity of different Cv-c protein variants to rescue the testis defects of *cv-c^C524* homozygous embryos (**Figure 4A**).

As mentioned above, StART domain mutations in DLC3 are the suspected cause of gonadal dysgenesis in human patients. In agreement with this, we found that expression of a *UAS-cv-c^ΔStART* construct generating a Cv-c protein lacking the StART domain, does not significantly normalize the testis defects (**Figure 4C, I**). In comparison, the expression of the wild-type Cv-c protein rescued the abnormal phenotypes in more than 50% of the testes (**Figure 4C, G**). Surprisingly, expression of a Cv-c mutant protein substituting a highly conserved Arginine into Glutamine residue that has been shown to block the GAP domain activity *in vitro* and the protein function *in vivo* (**Leonard et al., 1998**; **Sotillos et al., 2013**; **Sotillos et al., 2018**) rescued the gonadal phenotypes to a better extent than the wild-type protein (**Figure 4C, H**). This may be due to the overexpression of a functional GAP protein resulting toxic, not allowing to appreciate the full rescue of the StART-mediated function, a phenomenon that has been described previously (**Hendrick and Olayioye, 2019**; **Holeiter et al., 2012**). This does not happen in Cv-c^GAPmut nor in DLC3^WT, which may have a less efficient GAP function in a *Drosophila* environment than Cv-c.

Moreover, analysis of embryos homozygous for *cv-c^7*, an allele which carries that exact GAP mutation in the endogenous gene (**Denholm et al., 2005**), showed normal testis (**Figure 5A, B**), suggesting that Cv-c function in the gonad is not mediated through its RhoGAP function, but requires the StART domain, deleted in *cv-c^C524* (**Figure 5C, D**) and *cv-c^M62* mutant alleles (**Figure 5E, F**). To test if these results are due to Cv-c function in the testis, we repeated the experiments using *C587-Gal4*, a line expressed specifically in the gonad mesoderm cells ensheathing the germ cells. Although with a lower penetrance, this line results in similar rescues indicating a specific requirement of Cv-c in the testis gonadal mesoderm (**Figure 5—figure supplement 1**).

Although DLC3 and Cv-c are RhoGAP proteins that when mutant cause Rho1 over-activation, our experiments suggest that in the gonad these proteins do not require the GAP function. To confirm that the testis phenotype is not due to Rho1 over-activation we analysed if *cv-c^M62* gonad mutant phenotypes can be rescued by a *Rho1* mutation. As previously described, in *Rho1* homozygous mutants the gonad precursor germ cell migration is less efficient, giving rise to smaller male and female gonads (**Kunwar et al., 2003**). However, in *Rho1* mutants, the cells reaching the gonads become ensheathed and coalesce to form stable testes as in the wild type (**Figure 5G, H**). In *Rho1 cv-c* double mutant embryos, we observe an additive effect of both mutations (**Figure 5K, L**), with smaller gonads due to the *Rho1* early migratory defect and the testis germ cell extrusion typical of *cv-c* mutants (**Figure 5I, J**), indicating that the dysgenic gonad phenotype cannot be rescued by a reduction of Rho1 function.

These results show that the StART domain function is required for human and *Drosophila* testis formation strongly supporting that the dysgenic gonad defects observed in patients are caused by the DLC3-StART mutation and demonstrating that Cv-c has a GAP-independent function that requires the StART domain.

## Cellular causes for testis dysgenesis

We next searched for the underlying cellular defects responsible for the observed testis dysgenesis in *cv-c* mutants. To explore if the germ cell extrusion is due to a failure of the somatic gonadal cells to ensheath the germ cells, we studied the expression of E-cadherin (E-cad), which localizes to the membrane of both germ cells and somatic mesodermal cells and is required for germ cell ensheathment and gonadal compaction (**Jenkins et al., 2003**). We observe that while, in the wild-type testes the germ cells inside the gonad are encircled by high levels of E-cad protein (**Figure 6A, A'**), in *cv-c* mutants there are frequent gaps of E-cad distribution between adjacent germ cells inside the testes. In addition, we observe that extruded germ cells have almost no E-Cad on their membranes

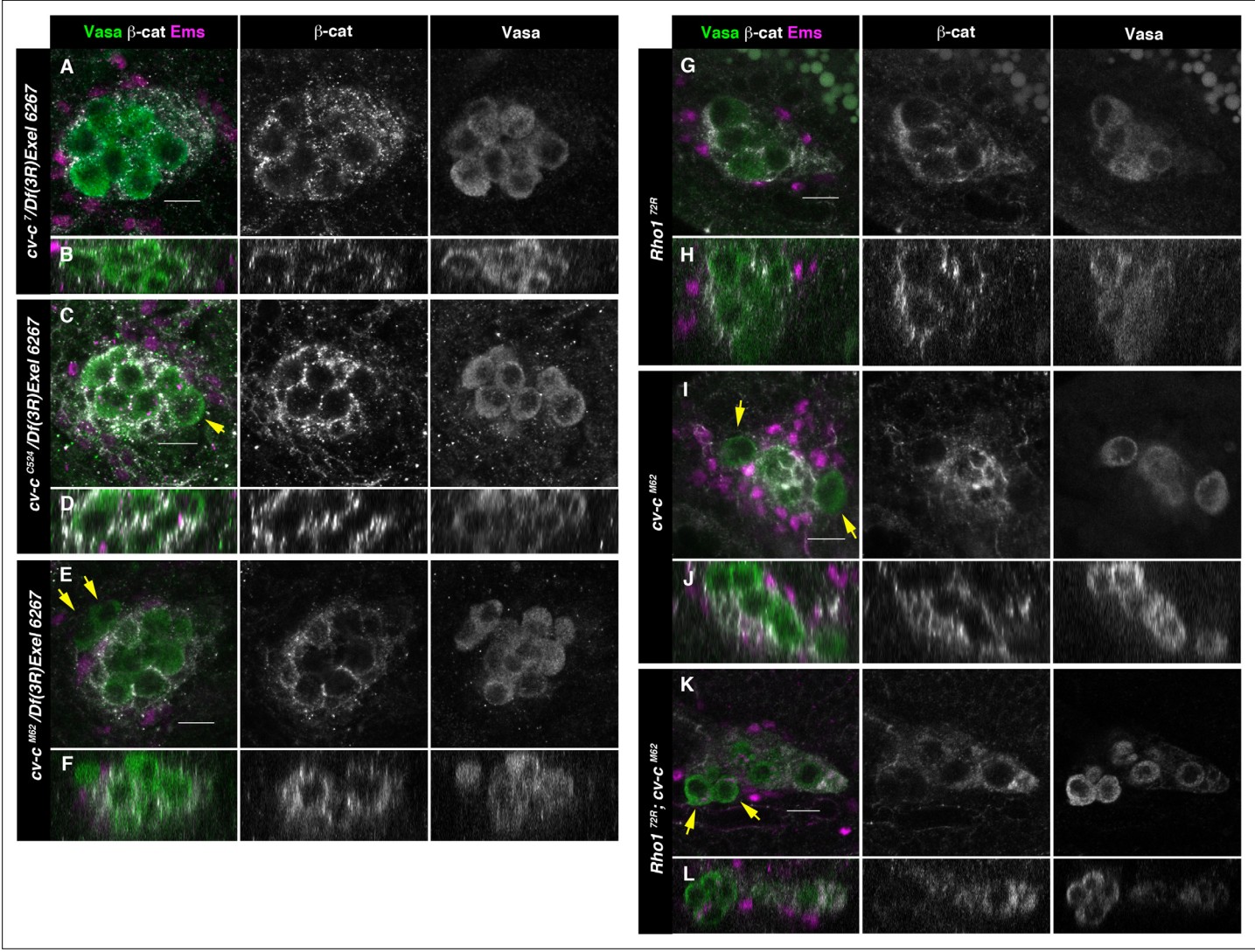

**Figure 5.** Testes in embryos with altered small GTPase regulation. (**A–L**) Testes of various genotypes stained with anti-Vasa (green), β-catenin (grey), and Ems (magenta). β-Catenin and Vasa channels are shown separately in right panels. (**A, B**) No extruded gem cells are observed in hemizygous *cv-c⁷/Df(3R)Exel6267* embryos carrying a *cv-c⁷* allele which inactivates the RhoGAP domain's function. (**C, D**) Hemizygous *cv-cᶜ⁵²⁴/Df(3R)Exel6267* or (**E, F**) *cv-cᴹ⁶²/Df(3R)Exel6267* showing extruded germ cells. (**G, H**) *Rho1⁷²ᴿ* mutant embryos have smaller testes without extruded germ cells. (**I, J**) Homozygous *cv-cᴹ⁶²* mutants present extruded germ cells. (**K, L**) Homozygous *Rho1⁷²ᴿ cv-cᴹ⁶²* double mutant embryos present smaller testis with extruded germ cells. Arrows point to GCs outside the gonads. Note that β-catenin envelops all germ cells in (**A, B**) and (**G, H**) while in (**C–F**) and (**I–L**) some germ cells are not surrounded (arrows). Z sections are shown under all panels. Scale bar: 10 µm.

The online version of this article includes the following source data and figure supplement(s) for figure 5:

**Figure supplement 1.** *cv-cᶜ⁵²⁴* homozygous mutant testes expressing different Cv-c and DLC3 protein variants exclusively in the somatic gonadal mesoderm with the *c587-Gal4* driver line.

**Figure supplement 1—source data 1.** Raw data.

(*Figure 6B, B'*, arrowheads). We observe analogous, abnormal β-catenin (*Figure 3*) localization inside the testes, indicating that the relationship between the germ cells and the surrounding somatic cells is not well established or poorly sustained (*Figure 3C, E, G*).

In addition to E-Cad, we also analysed Neurotactin (Nrt) expression that, in normal fly embryos, localizes to the cell extensions produced by the interstitial mesodermal cells that ensheath the germ cells (*Figure 6C, C'*; *Jenkins et al., 2003*). In *cv-c* mutant testes we find that Neurotactin expression is almost absent around the germ cells inside the gonad (*Figure 6D, D'*). Accordingly, interstitial somatic gonad mesodermal cells (labelled by the Traffic jam (Tj) antibody), which in the wild-type gonad can

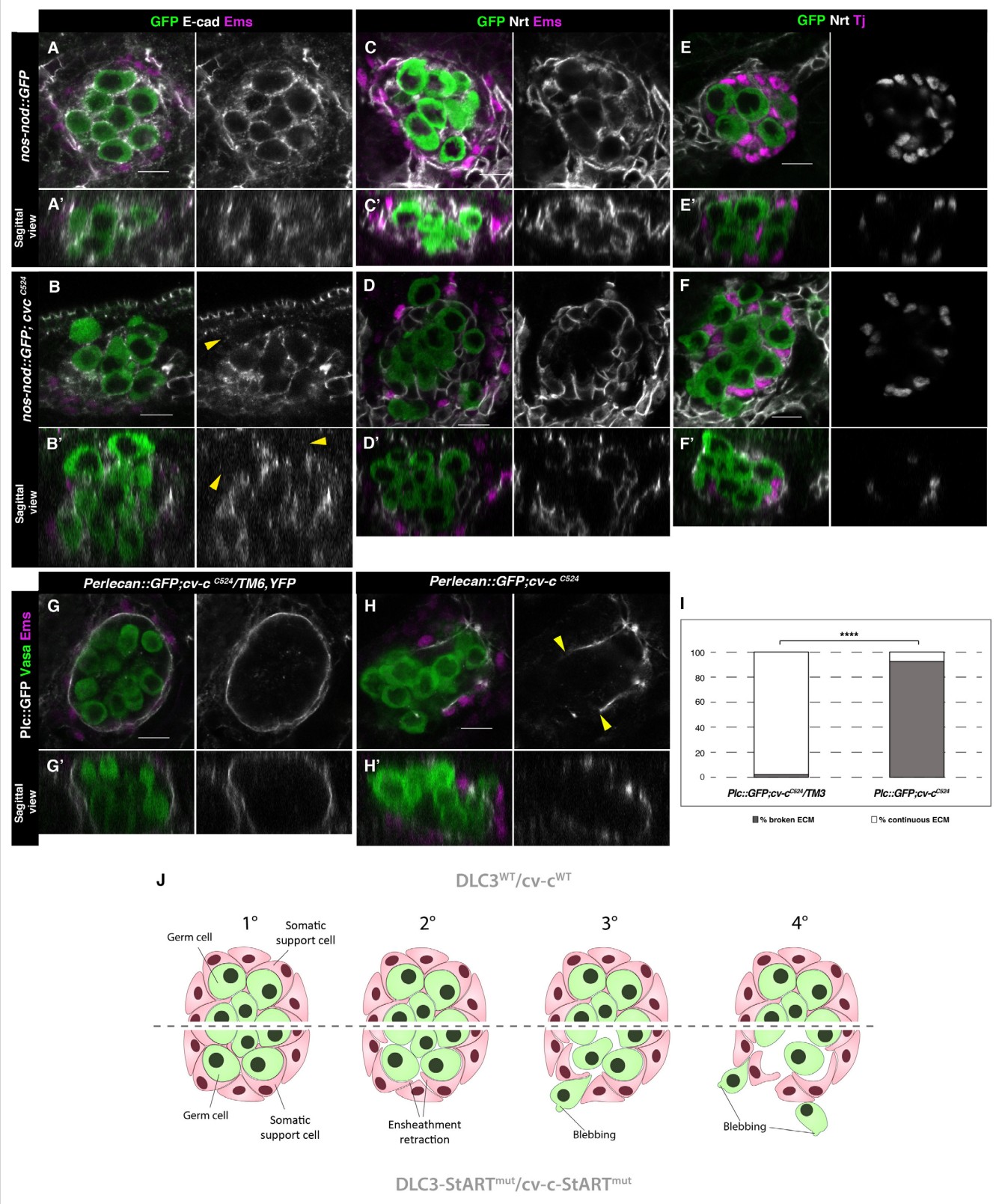

**Figure 6.** Ensheathment defects in *cv-c* mutant testes. (**A, C, E**) Wild-type and (**B, D, F**) *cv-c$^{C524}$* testes. Germ cells are labelled with *nos-nod::GFP* (green A–F) and labelled in grey with anti-E-cad (A–B) or anti-Nrt (C–D) to highlight the ensheathing membranes (grey in right panels). (**A, A'**) In the wild-type E-cad highlights contacts between the germ cells and the interstitial somatic gonadal cells surrounding each germ cell reflecting their correct ensheathment. (**B, B'**) In *cv-c* mutant testes, several germ cells inside the testis and all the extruded ones (arrowheads) are not surrounded by E-cad

*Figure 6 continued on next page*

*Figure 6 continued*

labelling membranes indicating incorrect ensheathment. (**C, C'**) Neurotactin in the wild-type testis reflects correct germ cell ensheathment. (**D, D'**) In *cv-c^C524* mutants little Neurotactin expression is detected inside the testis. Nuclei of Tj-labelled somatic gonadal cells (magenta and grey in right panels) are detected between the GCs inside wild-type testes (**E, E'**) but not in *cv-c^C524* mutant testes (**F, F'**). (**G, G'**) Heterozygous and (**H, H'**) homozygous *cv-c^C524* testes labelling the extracellular matrix with Perlecan::GFP (Pcl, grey in right panels) and stained with anti-Vasa (green) and anti-Ems (magenta) to label the GCs and the pigment cells, respectively. (**G, G'**) The wild-type testis is enclosed by a Perlecan containing extracellular matrix (white). In *cv-c^C524* mutant testis a discontinuous extracellular matrix is observed where the GCs are outside the gonad (H, H', yellow arrowheads). Z sections are shown under all panels. Scale bar: 10 µm. (**I**) Quantification of broken and continuous Perlecan Extracellular matrix (ECM) layer in *cv-c^C525* heterozygous or homozygous mutants using Fisher test; ****p value less than 0.0001 (*cv-c^C524*/TM6B N = 54 and *cv-c^C524* N = 24) (*Figure 6—source data 1*). (**J**) Interpretation of the testis degeneration in wild type and DLC3/Cv-c mutants. (**1**) The StART domain of DLC3/Cv-c has a Rho-independent function stabilizing cell interactions between germ cells and somatic support cells. Stable cell–cell interaction among the cells of the gonadal niche allows them to settle down in the testis. (**2**) In DLC3^S997N/Cv-c^ΔStART testis cell–cell interactions become compromised, cells lose their cohesion and separate from the gonadal niche. (**3**) After losing communication with somatic cells, germ cells become extruded from the gonadal niche initiating an erratic migrating behaviour. (**4**) Progressive loss of germ cells leads to gonad degeneration.

The online version of this article includes the following source data and figure supplement(s) for figure 6:

**Source data 1.** Raw data.

**Figure supplement 1.** ECM rescue of *cv-c* mutant testes.

**Figure supplement 1—source data 1.** Raw data.

be detected distributed between GCs (*Figure 6E, E'*), were frequently displaced to the periphery in *cv-c* mutant testes (*Figure 6F, F'*).

Finally, we studied the gonad integrity in *cv-c* mutants. Wild-type testes are surrounded by a Perlecan-rich extracellular matrix located between the pigment cells and the interstitial mesodermal cells that can be detected using a GFP insertion in the *Perlecan* gene (*Figure 6G, G', I*). In *cv-c^C524* mutants, the pigment cell layer and the ECM matrix are discontinuous with the extruded germ cells locating where the matrix gaps appeared (*Figure 6H, H'* arrowheads and **I**). These findings suggest that germ cell ensheathment by somatic cells cannot be maintained in the testis of *cv-c* mutant flies resulting in germ cell extrusion and gonadal rupture, a phenotype that is rescued by both Cv-c and DLC3 proteins containing a wild-type StART domain (*Figure 6—figure supplement 1*). Taken together our results show that the DLC3/Cv-c protein family has a StART-dependent function required for male gonadogenesis in humans and *Drosophila*.

## Discussion

Sex development is a central event in the life of metazoan animals. Sexual reproduction depends, particularly in mammals, on the determination of the gonads. What makes the gonads unique among all other organs is the fact that from one primordium two morphologically and functionally distinct organs, a testis and an ovary, arise. This phenomenon requires the fine tuning in space and time of all the genes, cells, and processes involved (*Carré and Greenfield, 2016*). Although new players are continuously discovered, many mechanisms underlying sexual determination and differentiation remain poorly understood, often due to the lack of a reliable experimental model. With this work, using *Drosophila* as a model, we confirmed the implication of *DLC3* as a novel DSD gene required for testis determination through the action of its StART domain.

*DLC3* had been found to be mutated in two 46,XY DSD siblings presenting gonadal dysgenesis but no experimental evidence confirming causality had been provided (*Ilaslan et al., 2018*). In this study, we report a third DSD patient with a mutation in DLC supporting the involvement of this gene in gonad development. First described in human myeloid cells, *DLC3* loss of expression was found in primary tumours from different tissues (*Durkin et al., 2007*). This multidomain protein forms part of a RhoGAP conserved family containing an N-terminus SAM domain followed by a serine-rich region, a catalytic GAP domain, and a StART domain (reviewed in *Braun and Olayioye, 2015*). There are also alternative isoforms lacking the SAM domain such as DLC3-β (*Durkin et al., 2007*). Despite the recent advances identifying its structure and spatial subcellular location (*Braun et al., 2015*; *Durkin et al., 2007*; *Hendrick et al., 2016*; *Holeiter et al., 2012*), the specific function of the various protein domains is still not completely understood. At the adherens junctions (AJ) of epithelial cells, DLC3-GAP domain inhibitory effect on RhoA stabilizes E-cadherin-based cell–cell contacts (*Hendrick*

*and Olayioye, 2019*). DLC3 is also known to influence the dynamics of the AJ indirectly, regulating E-cadherin turnover at the recycling compartments (*Braun et al., 2015*).

In search for the molecular elements mediating DLC3/Cv-c testis function, we studied the involvement of the StART and GAP domains. In the fly, Cv-c is expressed in the testis mesodermal cells during development and *cv-c* mutations deleting the GAP and the StART domains disrupt testis development without affecting ovarian development. Expression of the DLC3 human protein in *Drosophila* mutants can substantially rescue the male gonadal defects while the expression of the mutant DLC3$^{S993N}$ form found in DSD patients does not, proving a conserved functionality among both species and the implication of DLC3 as a novel DSD gene.

The only enzymatically active domain previously recognized in DLC3/Cv-c was the GAP domain with the StART domain thought to play a regulatory role. Surprisingly, the expression of Cv-c without a GAP-functional domain in a *cv-c* mutant background is able to rescue the testis' developmental disruption with a similar efficiency than wild-type DLC3 transgenic flies. However, transgenes without the StART domain (Cv-c$^{\Delta StART}$) or the human DLC3$^{S993N}$ allele carrying a mutation in the StART domain were unable to rescue the male gonad defects. These results demonstrate that the Cv-c StART domain has a GAP-independent function required for male gonad formation in *Drosophila*. StART domains have been shown to be lipid binding domains (*Gatta et al., 2018*; *Horenkamp et al., 2018*; *Iaea et al., 2015*; *Khelashvili et al., 2019*; *Naito et al., 2019*). The observation of a disrupted somatic-germ cell niche in the testis of *cv-c* mutant flies, made us wonder whether the StART domain's function in gonad development might be linked to lipid-regulated cell–cell interactions.

Our modelling analysis predicts that the S993N DLC3 mutation affects the Ω1-loop structure of the StART domain. Ω-loops play multiple roles in protein function, often related to ligand binding, stability, and folding (*Fetrow, 1995*). This loop is conserved in the StART domains of several other STARD/DLC proteins and seems to be functionally important and modulate access to the ligand-binding cavity (*Gatta et al., 2018*; *Gatta et al., 2015*; *Horenkamp et al., 2018*; *Naito et al., 2019*). Our in silico analysis of the DLC3-StART domain predicts the Ω1-loop displays the highest frequency of interaction with the membrane. In line with these, in silico simulation of the DLC3$^{S993N}$ mutant StART domain predicted the loss of one of the two conformational states of the protein, impairing its conformational dynamic with the loss of a closed state ligand-binding pocket in the StART domain. Within the loop, we find that residues S993, M994, A995, P996, and H997, have the highest frequency of interaction with the lipid bilayer. Interestingly, these five residues are conserved in the StART domains of StARD12 and StARD13, with the proline residue (P996 in DLC3) also conserved in StARD2, StARD7, StARD10, and StARD11 (*Thorsell et al., 2011*). These results suggest that the mutated S993N residue affects a critical structure of the protein domain and that alterations of this Ω1-loop could impair the domain's membrane-binding activity independently from the GAP domain. In this context, single mutations in the Ω1-loop surroundings, like the one carried by the patients, may have drastic effects on overall protein stability with consequences on the maintenance of the junctions between gonadal precursor cells.

In vertebrates and flies, gonad formation requires that germ cells migrate to the gonadal microenvironment, interact with the somatic cells, and cease the migrating behaviour. Germ cells migrate following an increasing gradient of chemo-attracting signals (*Richardson and Lehmann, 2010*). These signals trigger cell polarization in germ cells, reorganizing the membrane's cadherins. The protein rearrangement is followed by a reorganization of the cytoskeleton, creating a pulsation that flows front to back (*Kardash et al., 2010*; *Kunwar et al., 2008*; *Kunwar et al., 2003*). During this process there is a characteristic formation of 'blebs' or protrusions at the front of the migrating cell (*Kardash et al., 2010*). Germ cells migrate until they reach a point of maximum chemoattractant concentration. However, when subjected to ectopic signals, germ cells continue to show protrusions and migrating behaviour without a clear polarization (*Kardash et al., 2010*; *Richardson and Lehmann, 2010*). The PGCs' close association with the SGPs end the migratory phase, the last PGCs divisions are detected prior to compaction completion and the germ cells stop extending protrusions entering an 'inactive' phase (*Jaglarz and Howard, 1995*).

We show that in the absence of Cv-c function in the *Drosophila* testis, the mesodermal pigment cells do not form a continuous layer around the testis and the ECM surrounding the testis breaks. We also show that the interstitial gonadal cells fail to ensheath the germ cells and as a result of these defects the germ cells become extruded from the testis. These perturbations can be partially

corrected by expression in the testis mesoderm of human DLC3 or *Drosophila* Cv-c that in both cases require a functional StART domain. Thus, our results suggest that Cv-c/DLC3 have a fundamental function on the mesodermal testis cells that has been conserved. These results indicate that, as in *Drosophila*, the primary cause for the gonadal dysgenesis in DLC3 human patients is due to the abnormal maintenance of the testis mesoderm cells, which include both Sertoli and Leydig cells.

The germ and somatic gonadal cells interact via the formation of AJs in sex-specific patterns (*Fleming et al., 2012*). The junction protein complex in male gonads is relatively well conserved among species, with its core constituted by head-to-head cadherin dimers established from opposing interacting cells (*Piprek et al., 2020*; *Troyanovsky, 2012*). As in *Drosophila*, human PGCs have to migrate from the allantois to the genital ridge where the gonad is formed and later they interact with the Sertoli mesodermal cells. This interaction is essential for the PGCs to differentiate and survive and requires, both in humans and *Drosophila*, an E-cad levels increase when the PGCs and SGPs meet (*Fleming et al., 2012*; *Kunwar et al., 2008*). This initial interaction is fragile and must be stabilized by incorporating β-catenin and other proteins to the complex (*Fleming et al., 2012*; *Piprek et al., 2020*). The complex serves finally as an anchor for the actin filaments of the cytoskeleton. Accordingly, the alteration of E-cadherin and β-catenin distribution observed in *cv-c* and *DLC3* (humanized) fly mutant male gonads, together with the appearance of 'blebs', suggest that the settling-down switch has not been activated in the affected cells, and they remain in an erratic migrating behaviour until they eventually escape from the gonadal niche (summarized in *Figure 6J*).

During the compaction stage, the *Drosophila* somatic cell's E-cad locates into thin membrane extensions that surround each germ cell, which also expresses E-cad (*Jenkins et al., 2003*). In our study, we observed that E-cadherin and β-catenin were abnormally distributed in *cv-c* mutants, suggesting alterations of the connectivity between PGCs and SGPs. Interestingly, this phenomenon was observed even under Rho1 reduction, reinforcing the idea of a role of Cv-c in a Rho1-independent mechanism that promotes the stabilization of AJ after gonad coalescence.

This Rho-independent mechanism requires StART domain's integrity. Although most of the previous efforts to explain the molecular function of DLC3-StART domain have failed (*Alpy and Tomasetto, 2005*), work by *Sanchez-Solana et al., 2021* proved that the DLC1–3-StART domains bind phosphatidylserine (PS). They also postulated that, in DLC1, the lipid-binding works as a mediator of the interaction with several proteins independently from Rho-GAP domain activity (*Sanchez-Solana et al., 2021*). In eukaryotic membranes, PS is a well-known phospholipid involved in signalling pathways (*Kay and Grinstein, 2013*). In healthy cells, most PS locates on the inner layer of the plasma membrane. When the AJs are established, PS forms trans-bilayer-coupled nanoclusters with GPI-anchored proteins and glycosphingolipids from the outer layer. PS microdomains become anchoring points for proteins that promote signal transduction and the stabilization of the junction like Flotillins and PS-binding proteins (*Yap et al., 2015*). Taking these findings together, we hypothesize a role for the StART domain in the interaction of DLC3 with the temporal PS nanoclusters formed at the AJs, a phenomenon already demonstrated for DLC1 (*Sanchez-Solana et al., 2021*). Mutated DLC3-StART domain lacking the ability to respond to lipid binding with a conformational change, could be incapacitated to promote any further signalling.

The coincident requirement of DLC3/Cv-c for testis development and the conservation of the StART function suggest that in humans the DLC3-StART domain activity could also be required for the Sertoli cells/SGPs to trigger the germ cell settling behaviour and consolidate the male gonad development. It is still unknown why ovaries, where a similar somatic ensheathment of the germ cells occurs, do not require DLC3/Cv-c to maintain its stability. A possible explanation could be the sex-specific influence of AJ in overall patterning of the testis versus the ovary at the time of early gonadal sex differentiation (*Fleming et al., 2012*).

In conclusion, we demonstrate that mutations in DLC3/Cv-c are a novel cause of testicular dysgenesis in *Drosophila* and humans. Our results suggest that the dysgenesis is caused by the observed destabilization of cell–cell connexions between testis mesodermal cells and between the germ cells and somatic support cells after gonad coalescence. DLC3/Cv-c action in the gonads sheds new lights on the mechanism by which the germ cells end their migrating behaviour and settle in the gonadal niche. Pending the analysis of DLC3/Cv-c in further species, our results indicate this function could be a conserved mechanism among species, since human DLC3 is able to rescue the *cv-c* testicular defects in *Drosophila* embryos. Our work points to DLC3/Cv-c as a novel gene required specifically in

testis formation. Adding DLC3 to the list of genes involved in 46,XY complete dysgenesis opens up a new avenue to analyse the molecular and cellular mechanisms behind these disorders that could help in diagnosis and the development of future treatments. These results also underline the outstanding potential of *Drosophila* as a model to unveil the functional mechanism underlying human conditions like DSD without the ethical and logistical complexity of more conventional mammalian models.

## Acknowledgements

This work was supported by María de Maeztu Unit excellence grants MDM-2016-0687 and CEX-2020-001088 M and a Ministerio de Ciencia e Innovación grant PID2019-104656GB-I00 cofunded by the European Regional Development Fund (FEDER) to JC-GH. SV acknowledges support from the Swiss National Science Foundation (PP00P3_194807). This work was also supported by grants from the Swiss National Supercomputing Centre under project ID s1132 and has received funding from the European Research Council under the European Union's Horizon 2020 research and innovation program (grant agreement no. 803952). AL-B acknowledges support from the Swiss National Science Foundation's Grant 320030-184807. LL acknowledges support from the Swiss National Science Foundation (grant number SCOPES IZ73Z0_152347/1) and National Academy of Sciences of Ukraine, project 'Molecular-Genetic Mechanisms of Human Disorders of Sexual Development' [0121U110054]. We thank Acaimo González-Reyes for the critical reading of the manuscript and Steve DiNardo, Paul Lasko, Steve Russell, Uwe Walldorf, Acaimo González-Reyes, and Erika Matunis for generously sharing reagents. We also thank the Bloomington Stock Center and the Developmental Studies Hybridoma Bank.

## Additional information

### Funding

| Funder | Grant reference number | Author |
| --- | --- | --- |
| Maria de Maeztu Unit Excellence grants | MDM-2016-0687 | James Castelli-Gair Hombría |
| Maria de Maeztu Unit Excellence grants | CEX-2020-001088-M | James Castelli-Gair Hombría |
| Ministerio de Ciencia, Innovación y Universidades | PID2019-104656GB-I00 | James Castelli-Gair Hombría |
| Swiss National Science Foundation | PP00P3_194807 | Stefano Vanni |
| Swiss National Supercomputing Center | s1132 | Stefano Vanni |
| H2020 European Research Council | 803952 | Stefano Vanni |
| Swiss National Science Foundation | SCOPES IZ73Z0_152347/1 | Ludmila Livshits |
| National Academy of Sciences of Ukraine | 0121U110054 | Ludmila Livshits |
| Swiss National Science Foundation | 320030-184807 | Anna Biason-Lauber |

The funders had no role in study design, data collection, and interpretation, or the decision to submit the work for publication.

### Author contributions

Sol Sotillos, Conceptualization, Resources, Data curation, Formal analysis, Supervision, Investigation, Methodology, Writing – original draft, Writing – review and editing; Isabel von der Decken, Data curation, Investigation, Writing – review and editing; Ivan Domenech Mercadé, Investigation; Sriraksha Srinivasan, Data curation, Investigation; Dmytro Sirokha, Data curation; Ludmila Livshits, Resources,

Funding acquisition; Stefano Vanni, Software, Writing – review and editing; Serge Nef, Resources, Writing – review and editing; Anna Biason-Lauber, Conceptualization, Resources, Supervision, Funding acquisition, Writing – review and editing; Daniel Rodríguez Gutiérrez, Conceptualization, Resources, Writing – original draft, Writing – review and editing; James Castelli-Gair Hombría, Conceptualization, Resources, Funding acquisition, Investigation, Methodology, Writing – original draft, Writing – review and editing

## Author ORCIDs
Sol Sotillos ⓘ http://orcid.org/0000-0003-4731-8107
Ivan Domenech Mercadé ⓘ http://orcid.org/0000-0003-2115-8475
Stefano Vanni ⓘ http://orcid.org/0000-0003-2146-1140
Serge Nef ⓘ http://orcid.org/0000-0001-5462-0676
Anna Biason-Lauber ⓘ http://orcid.org/0000-0001-8966-2913
Daniel Rodríguez Gutiérrez ⓘ http://orcid.org/0000-0002-2979-0769
James Castelli-Gair Hombría ⓘ http://orcid.org/0000-0003-4183-5250

## Ethics
All clinical investigations were performed according to the declaration of Helsinki principles. The study was approved by the Geneva ethical committee CCER, authorization number 14-121. The patients and/or their legal guardians gave informed written consent to the study.

## Decision letter and Author response
Decision letter https://doi.org/10.7554/eLife.82343.sa1
Author response https://doi.org/10.7554/eLife.82343.sa2

## Additional files

### Supplementary files
• MDAR checklist
• Reporting standard 1. STROBE checklist.

### Data availability
All data generated during this study are included in the manuscript.

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

# Appendix 1

## Appendix 1—key resources table

| Reagent type (species) or resource | Designation | Source or reference | Identifiers | Additional information |
|---|---|---|---|---|
| Genetic reagent (*Drosophila melanogaster*) | *Rho1$^{72R}$* | | | *Strutt et al., 1997* |
| Genetic reagent (*Drosophila melanogaster*) | y[1]w[*];Mi{y[+mDint2]=MIC}cv-c[MI00245] | BDSC | RRID:BDSC_30677 | |
| Genetic reagent (*Drosophila melanogaster*) | P{Sxl-Pe-EGFP.G}G5b, w* | BDSC | RRID:BDSC_32565 | |
| Genetic reagent (*Drosophila melanogaster*) | w$^{1118}$; P{10XStat92E-GFP}1 | BDSC | RRID:BDSC_26197 | *Bach et al., 2007* |
| Genetic reagent (*Drosophila melanogaster*) | *cv-c$^{M62}$* | | | *Denholm et al., 2005* |
| Genetic reagent (*Drosophila melanogaster*) | *cv-c$^{7}$* | | | *Denholm et al., 2005* |
| Genetic reagent (*Drosophila melanogaster*) | *cv-c$^{C524}$* | | | *Denholm et al., 2005* |
| Genetic reagent (*Drosophila melanogaster*) | *UAS-cv-c$^{WT}$* | | | *Sotillos et al., 2018* |
| Genetic reagent (*Drosophila melanogaster*) | *UAS-cv-c$^{GAPmut}$* | | | *Sotillos et al., 2018* |
| Genetic reagent (*Drosophila melanogaster*) | *UAS-cv-c$^{\Delta StART}$* | | | *Sotillos et al., 2018* |
| Genetic reagent (*Drosophila melanogaster*) | *UAS-Myc-DLC3$^{WT}$* | | | *Sotillos et al., 2018* |
| Genetic reagent (*Drosophila melanogaster*) | *nos-nod::GFP* | A. González Reyes | | |
| Genetic reagent (*Drosophila melanogaster*) | *c587-Gal4* | E. Matunis | RRID:BDSC_67747 | |
| Genetic reagent (*Drosophila melanogaster*) | *six4-moe::GFP* | S. DiNardo | | *Sano et al., 2012* |
| Genetic reagent (*Drosophila melanogaster*) | *Vasa::GFP* | P. Lasko | | |
| Genetic reagent (*Drosophila melanogaster*) | *Perlecan::GFP* | | | *Morin et al., 2001* |
| Genetic reagent (*Drosophila melanogaster*) | *twist-G4* | | | *Greig and Akam, 1993* |
| Genetic reagent (*Drosophila melanogaster*) | *UAS-Myc-DLC3$^{S993N}$* | This paper | | Methods |

*Appendix 1 Continued on next page*

*Appendix 1 Continued*

| Reagent type (species) or resource | Designation | Source or reference | Identifiers | Additional information |
|---|---|---|---|---|
| Recombinant DNA reagent | XhoI DLC3$^{WT}$::pBS | This paper | | Methods |
| Recombinant DNA reagent | XhoI DLC3 $^{S993N}$::pBS | This paper | | Methods |
| Recombinant DNA reagent | pUASt::Myc-DLC3$^{S993N}$ | This paper | | Methods |
| Sequence-based reagent | DLC3S993N-For | This paper (Sigma-Aldrich) | PCR primers | 5'-TGTACCACTATGTCAC CGACA-A-CATGGCACC-3' |
| Sequence-based reagent | DLC3S993N-Rev | This paper (Sigma-Aldrich) Methods | PCR primers | 5'-TGGGGTGCCATG-T-TGTCGGTGACATAGTG-3' |
| Antibody | Rat monoclonal anti-VASA | DSHB | AB_760351 | (1:20) |
| Antibody | Mouse monoclonal anti-Nrt | DSHB | AB_528404 | (1:100) |
| Antibody | rat monoclonal anti-DE-cad | DSHB | AB_528120 | (1:50) |
| Antibody | Rabbit polyclonal anti-STAT | E. Bach | *Flaherty et al., 2010* | (1:500) |
| Antibody | Mouse monoclonal anti-β-catenin | DSHB | AB_528089 | (1:100) |
| Antibody | mouse monoclonal anti-Eya | DSHB | AB_528232 | (1:100) |
| Antibody | guinea pig polyclonal anti-Ems | U. Walldorf | *Walldorf and Gehring, 1992* | (1:5.000) |
| Antibody | Rabbit polyclonal anti-Sox100B | S. Russell | *Nanda et al., 2009* | (1:1000) |
| Antibody | guinea pig polyclonal anti-Tj | A. González-Reyes | *Díaz-Torres et al., 2021* | (1:1000) |
| Antibody | rabbit polyclonal anti-Perlecan | A. González-Reyes | *Díaz-Torres et al., 2021* | (1:1000) |
| Antibody | Rabbit polyclonal anti-GFP | Invitrogen | A11122 | (1:300) |
| Antibody | Chicken polyclonal anti-GFP | Abcam | ab13970 | (1:500) |
| Antibody | mouse anti-βgal | Promega | Z378A | (1:1000) |
| Antibody | Goat polyclonal anti-Mouse IgG (H+L) Highly Cross-Adsorbed Secondary Antibody, Alexa Fluor 488 | Invitrogen | A-11029 | (1:400) |
| Antibody | Goat polyclonal anti-Mouse IgG (H+L) Highly Cross-Adsorbed Secondary Antibody, Alexa Fluor 555 | Invitrogen | A-21424 | (1:400) |
| Antibody | Goat polyclonal anti-Mouse IgG (H+L) Highly Cross-Adsorbed Secondary Antibody, Alexa Fluor 647 | Invitrogen | A-21236 | (1:400) |
| Antibody | Goat polyclonal anti-Rabbit IgG (H+L) Highly Cross-Adsorbed Secondary Antibody, Alexa Fluor 488 | Invitrogen | A-11034 | (1:400) |
| Antibody | Goat polyclonal anti-Rabbit IgG (H+L) Highly Cross-Adsorbed Secondary Antibody, Alexa Fluor 555 | Invitrogen | A-21429 | (1:400) |
| Antibody | Goat polyclonal anti-Rabbit IgG (H+L) Highly Cross-Adsorbed Secondary Antibody, Alexa Fluor 647 | Invitrogen | A-21245 | (1:400) |
| Antibody | Goat polyclonal anti-Guinea Pig IgG (H+L) Highly Cross-Adsorbed Secondary Antibody, Alexa Fluor 555 | Invitrogen | A-21435 | (1:400) |

*Appendix 1 Continued on next page*

*Appendix 1 Continued*

| Reagent type (species) or resource | Designation | Source or reference | Identifiers | Additional information |
|---|---|---|---|---|
| Antibody | Goat polyclonal anti-Guinea Pig IgG (H+L) Highly Cross-Adsorbed Secondary Antibody, Alexa Fluor 647 | Invitrogen | A-21450 | (1:400) |
| Antibody | Goat polyclonal anti-Rat IgG (H+L) Highly Cross-Adsorbed Secondary Antibody, Alexa Fluor Plus 488 | Invitrogen | A-48262 | (1:400) |
| Antibody | Goat polyclonal anti-Rat IgG (H+L) Highly Cross-Adsorbed Secondary Antibody, Alexa Fluor Plus 555 | Invitrogen | A-48263 | (1:400) |
| Antibody | Goat polyclonal anti-Chicken IgY (H+L) Cross-Adsorbed Secondary Antibody, Alexa Fluor Plus 488 | Invitrogen | A32931 | (1:400) |
| Antibody | Donkey polyclonal anti-Goat IgG (H+L) Cross-Adsorbed Secondary Antibody, Alexa Fluor 555 | Invitrogen | A-21432 | (1:400) |
| Software | ImageJ/Fiji | Fiji | http://fiji.sc/ | |
| Software | AdobePhotoshop/Illustrator | Adobe | https://www.adobe.com/ | |
| Software | Prism | GraphPad | https://www.graphpad.com/data-analysis-resource-center/ | |
| Software | Imaris | Oxford Instruments | https://imaris.oxinst.com/ | |
| Software | SeqBuilder/SeqMan | DNASTAR | https://www.dnastar.com/ | |
| Commercial assay, kit | DIG RNA Labelling Kit | Roche | 11 175 025 910 | |
| Commercial assay, kit | Qiagen Plasmid Midi Purification Kit | Qiagen | 12143 | |
| Other | Rhodamine Phalloidin (1:100) | Invitrogen | R415 | High-affinity F-actin probe conjugated to red fluorescence dye TRITC |
| Other | VECTASHIELD Mounting Medium | Vector Laboratories | âH-1000 | Antifade Mounting Medium for preserving fluorescence |
| Other | DAPI (1:10.000) | Thermo Fisher Scientific | 62248 | Blue-fluorescent DNA stain |
| Other | Pfu polymerase | Promega | M774A | PCR polymerase |
| Other | DpnI restriction enzyme | Roche | 10742988001 | DNA restriction enzyme |
| Other | Halocarbon oil 27 | Sigma | H8773 | Inert oil to avoid *Drosophila* embryos/tissues desiccation. |

