## [Editor Report]

This important study demonstrates a conserved function of somatic human DCL3/ *Drosophila* Cv-c in the development of the testis. The evidence supporting the claims of the authors is compelling as they combine the human genetics of a patient with gonadal dysgenesis with genetic rescues in *Drosophila*. This work will be of interest to developmental biologists and human geneticists. A major strength of the paper is proving that the presumptive lipid-interacting StART domain (but not the GAP domain) of DCL3/Cv-c is required for testis development, which opens a new area of investigation.

---

## [Decision Letter]

[Editors' note: this paper was reviewed by Review Commons.]

---

## [Author Response]

General Statements

Our work shows the involvement of the Human DLC3 protein and its *Drosophila* ortholog Cv-c in testis development. This is a previously undescribed function for both of them, which *per se* is relevant. In addition, we show that DLC3 mutations in the StART domain region are associated with sexual dysgenesis at least in 3 patients. Thus, understanding of the regulation of DLC3/STARD8 genes, and what might perturb their function would appear to present a whole new area for exploration in relation to testicular dysgenesis. Our study is also relevant because we demonstrate that DLC3/Cv-c function in testis formation is independent of the GAP domain, and thus of Rho regulation, but depends on the StART domain. This is an unexpected finding for a RhoGAP protein and opens up a new avenue of research that should be addressed in future works.

Point-by-point description of the revisions

In what follows we describe how we have modified the reviewed manuscript.

Reviewer 1:Major comments:In general, the data support the conclusions. I cannot comment on the atomistic simulation experiment as it is outside of my expertise. I had some difficulties interpreting Figure 2 as the contrast in the colour panels made it difficult to assess the different staining patterns. I would recommend changing the blue to cyan for easier visibility. While I agree that there are some differences between Figure 2F and Figure 2G it is not simple for the non-expert to distinguish the gonadal mesoderm from the somatic mesoderm. I think the enlarged panels could do with also showing the overlap in staining, or at least a tracing of the different cell populations so that the gonadal mesoderm can be clearly defined. Please also add some scale bars to the figure. Figure 3 demonstrates clear differences in gonad morphology between male and female mutants but the contrast in the colour panels A-G could also be improved. Panels H-J are very clear.

As suggested by Referees 1 and 3 we have modified the colour channels in all figures. We have also enlarged the figures taking away the uninformative region and focused around the enlarged gonads and added scale bars. For Figure 2F-G, we have added a close up of the region of interest both in colour and in black and white. These changes have increased the contrast and facilitate the data interpretation to non-expert readers.

The rescue experiment in Figure 4 is clearly presented but could the DLC3 mutants in the graph (panel b) please be named similarly to the schematic proteins shown in panel a.

We have changed the names to maintain nomenclature uniformity.

I found the difference between the RhoGAP domain mutants and the StART domain mutants of Cv-c to be clearly defined, and correlate with DLC3 function. This is a very interesting result that indicates multiple molecular functions for the Cv-c /DLC family.The methods are well described, statistics adequate and the data well described.Minor Comments:My only suggestion for the text is to provide a more through description of the StART domain in the introduction.

We have included in the introduction the following paragraph describing the StART domain:

“This family of proteins share different domains: besides the Rho GTPase Activating Protein domain (GAP), they present a protein-protein interacting Sterile Α Motif (SAM) at the N terminal end and a Steroidogenic Acute Regulatory protein (StAR)-related lipid transfer (StART) domain at the C terminal. StART domains have been shown in other proteins to be involved in lipid interaction, protein localization and function.”

Reviewer 2:My only issue with the present study is to how well the present experimental findings in *Drosophila* translate to humans. As far as I can tell the present studies show that inactivating mutations in Cv-c in *Drosophila* result in failure of germ cell enclosure by somatic cells into the testis, resulting in sterility. In humans, and in experimental mouse transgenic lines, it has been well established that absence of germ cells does not of itself lead to failure of testis differentiation and onward development, nor does it lead automatically to sex reversal or impairment of masculinization. For the latter to occur, there must be impairment/failure of fetal Leydig cell function such that insufficient androgen is produced to effect genital/bodywide masculinization. Obviously, this will happen if no testis forms as appears to be the case in the new human DLC3 mutant reported in the present manuscript (although detail on this is unfortunately lacking). This appears to be different to the previous published DLC3/STARD8 mutant sisters, in whom the phenotype appears to reflect failure of steroidogenesis. Is the proposal that DLC3/STARD8 plays a role in both testis differentiation and in Leydig cell function (steroidogenesis) or is this due to different DLC3 genes? I think the authors need to address these key issues in their discussion, if only to highlight that there are at present many gaps in our understanding.

The reviewer says:

“As far as I can tell the present studies show that inactivating mutations in Cv-c in *Drosophila* result in failure of germ cell enclosure by somatic cells into the testis, resulting in sterility.”

This sentence does not represent the spirit of our findings accurately and this probably reflects the fact that we stressed the interaction between somatic mesodermal cells and germ cells in *Drosophila* which probably concealed that the main defects in *cv-c* mutants are caused by the abnormal interaction of the mesodermal cells with germ cells but also among themselves. Our study provides insights about a new conserved pathway required in the mesodermal cells for the maintenance of an already formed testis, and only indirectly can be considered to deal with sterility. We show that Cv-c is required in the mesodermal cells for the correct maintenance of the testis structure, that when it fails leads to the testis dysgenesis which, among other defects, releases the germ cells. We show that in the absence of Cv-c function in the testis, the mesodermal pigment cells do not form a continuous layer around the testis and the ECM surrounding the testis breaks. We also show that the interstitial gonadal cells fail to ensheath the germ cells and as a result of all these the germ cells become dispersed. These perturbations can be partially corrected by expression in the testis mesoderm of human DLC3 or *Drosophila* Cv-c that in both cases require a functional StART domain. Thus, our results suggest that Cv-c/DLC3 have a fundamental function on the mesodermal testis cells that has been conserved. These results indicate that, as in *Drosophila*, the primary cause for the gonadal dysgenesis in DLC3 human patients is due to the abnormal maintenance of the testis mesoderm cells, which include both Sertoli and Leydig cells. Thus, our proposal is that DLC3/STARD8 plays a role in testis maintenance through its function in mesodermal cells which will probably affect both Sertoli and Leydig cell function.

To clarify the issue raised by the Reviewer we have modified both the introduction and the discussion to highlight that although humans and *Drosophila* diverged more than 500 million years ago there are similarities regarding gonad stabilisation.

We have modified the introduction to clarify this issue:

“Gonadogenesis can be subdivided into three stages: specification of precursor germ cells, directional migration towards the somatic gonadal precursors and gonad compaction. In mammals, somatic cells, i.e. Sertoli cells in male and Granulosa cells in females, play a central role in sex determination with the germ cells differentiating into sperms or oocytes depending on their somatic mesoderm environment. In humans, Primordial Germ Cells (PGCs) are formed near the allantois during gastrulation around the 4th gestational week (GW) and migrate to the genital ridge where they form the anlage necessary for gonadal development (GW5-6). Somatic mesodermal cells are required for both PGC migration and the formation of a proper gonad. Once PGCs reach their destination, the mesoderm gonadal cells join them (around GW 7-8 in males, GW10 in females) and provide a suitable environment for survival and self-renewal until gamete differentiation {Jemc, 2011 #413}. Thus, mutations in genes regulating somatic Sertoli and Granulosa support cell function in humans are often associated with complete or partial gonadal dysgenesis in both sexes and sex reversal in males {Zarkower, 2021 #430; Knower, 2011 #418; Brunello, 2021 #399}. Other mesodermal cells, the Leydig cells, also play an important role in the testis by being the primary source of testosterone and other androgens and maintaining secondary sexual characteristics.”

Also, we have added a paragraph in the discussion to emphasize this argument:

“We show that in the absence of Cv-c function in the *Drosophila* testis, the mesodermal pigment cells do not form a continuous layer around the testis and the ECM surrounding the testis breaks. We also show that the interstitial gonadal cells fail to ensheath the germ cells and as a result of these defects the germ cells become extruded from the testis. These perturbations can be partially corrected by expression in the testis mesoderm of human DLC3 or Drosophila Cv-c that in both cases require a functional StART domain. Thus, our results suggest that Cv-c/DLC3 have a fundamental function on the mesodermal testis cells that has been conserved. These results indicate that, as in Drosophila, the primary cause for the gonadal dysgenesis in DLC3 human patients is due to the abnormal maintenance of the testis mesoderm cells, which include both Sertoli and Leydig cells”.

On the comment: “*Obviously, this will happen if no testis forms as appears to be the case in the new human DLC3 mutant reported in the present manuscript (although detail on this is unfortunately lacking).”*

We have added details on this new patient in the Results section: “The patient presented 46,XY gonadal dysgenesis, where the gonadectomies found on the right gonad connective tissue of the fallopian tube with thickened walls and partially obliterated lumen and, in some areas, fragments of ovarian tissue with sclerosed theca tissue and on the left gonad degenerative altered tissue of the testicle with areas of mucous and degeneration of tubules without signs of spermatogenesis.”

We also described the methodology in the Materials and methods section.

As a result of this addition to the text, we have included two additional authors (Dmytro Sirokha, Ludmila Livshits from the National Academy of Sciences of Ukraine, Kyiv)

that were directly involved in the patient analysis.

I would also suggest that the authors highlight another potentially more important spin-off from such studies, namely that understanding of the regulation of DLC3/STARD8 genes, and what might perturb their expression/action would appear to present a whole new area for exploration in relation to testicular dysgenesis/masculinization disorders.

We have modified the last part of the discussion to introduce Reviewer’s 2 suggestion:

“Our work points to DLC3/Cv-c as a novel gene required specifically in testis formation. Adding *DLC3* to the list of genes involved in 46X,Y complete dysgenesis opens up a new avenue to analyse the molecular and cellular mechanisms behind these disorders that could help in diagnosis and the development of future treatments”.

Reviewer 3:Major comments:1. This study has shown the expression pattern of cv-c and the consequence of cv-c mutation on different aspects of gonad development. However, one major comment is there is no quantification of the expression levels as well as the scoring of the mutant phenotypes.2. In Figure 2, for instance, I recommend that the authors display the quantification of the fluorescence intensity of the cv-c expression under all circumstances (in situ hybridization as well as protein-trap based GFP expression) to better depict the differences among the male vs female gonad.

As suggested by the reviewer, we have quantified all mutant phenotypes (see below). We have also increased the images’ contrast and size. We believe that the changes performed to the figures are sufficient to illustrate our statement about *cv-c* being expressed in testis but not in ovaries. We have not quantified *cv-c*’s RNA or protein levels as these will not add much to the results.

3. In Figure 3, the authors show the different gonad developmental defects associated with the cv-c mutation. Specifically, the authors show that the gonad mesoderm cells are displaced with the pigment cells failing to ensheath the germ cells. In addition, the authors also suggest that there is an increased frequency of germ cell blebbing, an indication of migratory activity. However, there is no quantification of these findings. I think the authors should display a quantitative estimation of % of the mutant gonad depicting these phenotypes vs the normal gonad to have a perspective of how penetrant the phenotypes are.

As suggested by the Reviewer, we have quantified the blebbing phenotype. The results are presented in figure 3, panel J.

4. In Figure 4, the authors attempt to rescue the Cv-C mutation linked gonadal defects by overexpressing different Cv-C protein variants. The rescue experiments are not very clear. The graph shows the % of normal testes under different genotypic combinations. It is not very clear what the authors mean by normal (in what context)? Since the mutation results in different defects of gonad development, I think recommend that represent the rescue in terms of these defects. It would be interesting to see for instance, what happens to the blebbing or germ cell ensheathment phenoype upon rescue. How many % of testes show the rescue as compared to cv-c mutants?

The percentages are quantified considering if the testes have any germ cell outside the gonad. We have added a line to clarify this point in the figure legend: *“…quantified as encapsulated gonads with all germ cells inside the testis as assessed by Fisher-test*”.

Furthermore, we have quantified the number of gonads with ECM breaks, using a specific antibody against Perlecan, obtaining similar results to gonads with extruded GCs. These results are shown in a new supplementary figure 5 (Figure 6—figure supplement 1), including the quantifications as well as representative images of the phenotypes.

5. Did the authors try cell-specific depletion of cv-c and examined the consequence on gonad development?

*cv-c* mutants are embryonic lethal because of Cv-c’s widespread requirement on various embryonic tissues during development. Induction of FRT clones in the embryonic testis mesoderm was unsuccessful because of the low number of divisions during embryogenesis. We also tried to knock down *cv-c* expression with 3 different RNAi lines. Unfortunately, overexpression of these RNAi with different testis Gal4 drivers did not decrease *cv-c* mRNA levels significantly in the mesoderm nor in other tissues where *cv-c* is expressed. Despite the unsatisfactory outcome of these experiments, our finding that *cv-c* is expressed in the testis mesodermal cells, and the fact that we can rescue the testis phenotypes by expressing Cv-c with gonadal mesodermal specific Gal4 lines supports a testis mesoderm requirement of *cv-c* for its gonadal function.

6. Another major concern is the lack of mechanistic insight of cv-c. For example, how does loss of cv-c result in gonadal dysgenesis? The authors suggested that StART domains regulate via lipid binding. The authors could examine if StART domain function is dependent on lipid-mediated interactions.

We agree with the Reviewer that the molecular characterisation of the StART-mediated GAP-independent Cv-c function we have uncovered in this work is a very interesting finding that should be addressed by future work. However, such biochemical characterisation falls beyond this work as it requires a complex approach to distinguish between the previously described StART function regulating the GAP activity (Sotillos et al. Scientific Reports 2018) from the new GAP-independent testes-specific function here we describe.

The central point of this manuscript is the demonstration that both DLC3 and Cv-c are required in testis, an important conserved function that had been overlooked by previous publications. Thus, DLC3 should be considered a new gene to be analysed in the future when studying gonadal dysgenesis. A second important point raised by our work is the demonstration that DLC3/Cv-c can perform RhoGAP independent functions, something that had never been described for these proteins.

Notwithstanding this, in the revised version, we have added a new supplementary figure (Figure 1—figure supplement 1) analysing *in-silico* the StART domain-lipid interaction. In-silico modelling shows that the DLC3-StART domain Ω1-loop structure displays the highest frequency of interaction with the membrane. This loop is conserved in the StART domains of several other STARD proteins and seems to modulate access to the ligand binding cavity. Ω-loops play multiple roles in protein function, often related to ligand binding, stability and folding. In this context, mutations in the proximity of the Ω1-loop, like the ones carried by the patients, may have drastic effects on overall protein stability that could affect the interaction between gonadal precursor cells.

7. Do the cv-c mutants survive to adulthood? If yes, then it would be interesting to know how the adult testis behaves in cv-c mutants. Does it result in sterility?

Unfortunately, all studied *cv-c* mutants are embryonic lethal.

8. Ensheathment is required for proper germline development and defects in ensheathment can affect soma-germline communication and germline development. Germ cell ensheathment affects the proliferation of germ cells and display defective JAK/STAT signaling. It would be interesting to know if the germ cells in cv-c mutant gonad show the proliferation defect and impaired JAK/STAT signaling.

This is an interesting suggestion as JAK/STAT signalling has a male specific function that could explain why *cv-c* gonadal defects are male specific (Sheng et al. Dev. Biol. 2009). To test any interaction between Cv-c and JAK/STAT signalling we have analysed if in *Df(1)os1A* mutant embryos that lack all three Upd *Drosophila* JAK/STAT ligands thus rendering an inactive pathway (Hombría et al. Dev. Biol.2005), the expression of cvc::GFP is affected. *Df(1)os1A/+; cvc::GFP/+* females were crossed to *cvc::GFP/+* males allowing us to compare Cvc::GFP expression in *Df(1)os1A* male embryos with their control siblings (Figure 2—figure supplement 1). Although, as previously described, the *Df(1)os1A* testis were smaller due to the lack of JAK/STAT activation, normal expression levels of Cvc::GFP were detected in the gonadal mesoderm.

We also analysed if Cv-c controls the activity of JAK/STAT pathway. First, we analysed, using anti-STAT, if STAT’s nuclear accumulation in *cv-c^C524^* mutant embryos is altered. Second, we analysed if Cv-c regulates the JAK/STAT pathway signalling using the 10XSTAT-GFP reporter which reveals the pathway’s activation (Bach, Gene Expr. Patterns. 2007). In both cases we did not find any difference between the heterozygous and homozygous *cv-c^C524^* testis (Sup Figure 2), indicating JAK/STAT and Cv-c phenotypes are independent of each other.

9. I was also wondering if the authors have examined the number of germ cells in the mutant gonads.

Yes, we have counted the number of germ cells in *cv-c* mutants and, if anything, there are a few more. We initially considered that an excess of germ cell proliferation could be the cause of gonad disruption. However, we have discarded this hypothesis as phospho-histone 3 staining did not show a significant increase of germ cell divisions. Moreover, blocking cell proliferation in *cv-c* mutant gonads using UAS-p21, does not rescue the testes phenotype. We are unsure what may cause the slight increase of germ cells observed.

10. In addition, I think the quality of the images should be improved.

We have changed the colours used in the confocal images and amplified the relevant regions in all panels. We thank both referees for this suggestion as these changes have improved the figure contrast.

Minor comments:1. cv-c mRNA in Figure 2 panels (Figure 2D) should be in italics.

We have changed it.

2. There is no scale bar in Figure panels. In addition, there is no scale bar in the zoomed images in Figure 2. Scale bars should be consistently put in the all the Figures, in particular on the first panels of the Figures.

We have added scale bars to all panels.

3. In the line 677, the manuscript says "arrowhead". There are no arrowheads but the arrows.

Corrected

4. Please be consistent with the labels in Figure panels: Vasa is shown in capital while Eya is not.

Corrected

5. Please be consistent with the labeling of the Figure panels: Figure 3A vs Figure 4a.

Corrected

6. What does the asterisk signify in Figure 2? There is no mention of asterisk in the Figure 2 legend.

The meaning of the asterisk is explained in the figure legend.

7. There is no grey channel (sagittal view) for the panels Figure 3I and J.

We have included sagittal views in the figure.

8. Please be thorough in labeling the genotypes in Figures. For instance, Figure 4c depict the % of normal testis in cv-c δ StART. However, the correct genotype is twi>Cv-c StART. In addition, in Figure 4c graph, cv-c mut should be cv-cGAPmut.9. Please be consistent with the depiction of the "START" domain of the protein throughout the manuscript. In figure 4c for instance, it is "START" in the graph while in the figure panel 4i, it is StART.10. In Figure 4b, it is written DLC3-GA. Did the authors mean DLC3-S993N?11. In line 723, it should be anti-β catenin.

As suggested, we have unified figure labelling.

12. The authors have shown two images to suggest that cv-c mutant gonad depict the germ cell blebbing (Figure 3I and J). I think it would be much better to put up a graph showing the number or percentage of cv-c mutant gonads displaying the germ cell blebbing than putting two images with the same information.

We have quantified the data which is included as a graph in figure (3J).

13. The previous comment is also true for Figure 6H and I. In both the panels, the authors wish to show discontinuous ECM marked by Perlecan expression in cv-c mutant gonads. I think it would be better to display a score of the number of mutant gonads depicting the discontinuous ECM.

To quantify Perlecan disruption in *cv-c* mutants we have repeated the stainings. The results are displayed as a graph in Figure 6 and as Figure6—figure supplement 1.